# KEVIN: MULTI-TURN RL FOR GENERATING CUDA KERNELS

**Carlo Baronio**[*]
Stanford University
Cognition AI
carlo@cognition.ai

**Pietro Marsella**[*]
Stanford University
marsella@stanford.edu

**Ben Pan**[*]
Stanford University
Cognition AI
ben@cognition.ai

**Simon Guo**
Stanford University
simonguo@stanford.edu

**Silas Alberti**
Cognition AI
silas@cognition.ai

## ABSTRACT

Writing GPU kernels is a challenging task and critical for AI systems' efficiency. It is also highly iterative: domain experts write code and improve performance through execution feedback. Moreover, it presents verifiable rewards like correctness and speedup, making it a natural environment to apply Reinforcement Learning (RL). To explicitly incorporate the iterative nature of this process into training, we develop a flexible multi-turn RL recipe that addresses unique challenges encountered in real-world settings, such as learning from long trajectories and effective reward attribution across turns. We present Kevin - K(ernel D)evin, the first model trained with multi-turn RL for CUDA kernel generation and optimization. In our evaluation setup, Kevin shows significant gains over its base model (QwQ-32B), improving correctness of generated kernels (in pure CUDA) from 56% to 82% and mean speedup from 0.53x to 1.10x of baseline (PyTorch Eager), and surpassing frontier models like o4-mini (0.78x). Finally, we study its behavior across test-time scaling axes: we found scaling serial refinement more beneficial than parallel sampling. In particular, when given more refinement turns, Kevin shows a higher rate of improvement.

## 1 INTRODUCTION

Writing efficient GPU kernels (Dao et al., 2022; Zhao et al., 2025; Ye et al., 2025) in domain-specific languages: CUDA, Triton, ThunderKittens, CUTLASS, etc. (Nickolls et al., 2008; Tillet et al., 2019; Spector et al., 2024; NVIDIA Corporation, 2025) is critical for enabling AI systems' efficiency at scale, yet it remains difficult and costly due to the deep domain expertise required. This has led to a surge of interest in exploring how Large Language Models (LLMs) could help generate GPU kernels (Ouyang et al., 2025; Li et al., 2025; NVIDIA, 2025) using agentic systems (Damani et al., 2024; Chen et al., 2025; METR, 2025; Lange et al., 2025; Google DeepMind, 2025) that leverage extensive test-time compute. These inference-based approaches are inherently limited by the base models' capability in this domain. On the other hand, the presence of verifiable rewards in the form of correctness and speedup against a reference implementation makes reinforcement learning (RL) a natural approach. This leads to our investigation: *How can we train a model using RL to solve the real-world engineering task of CUDA kernel generation?*

GPU kernel generation emphasizes not just functional correctness, but more importantly performance — distinguishing this code optimization problem from binary-reward tasks that involve passing unit tests (Jimenez et al., 2024) or producing an acceptable proof (Zheng et al., 2022). Since speedup is a continuous goal, performance engineers take an iterative approach: they conduct many rounds of optimization based on previous kernel code, its execution result, and timing profiles. Hence, arriving at an optimized solution relies on multiple turns conditioned on previous execution feedback. In

---

[*]Equal contribution

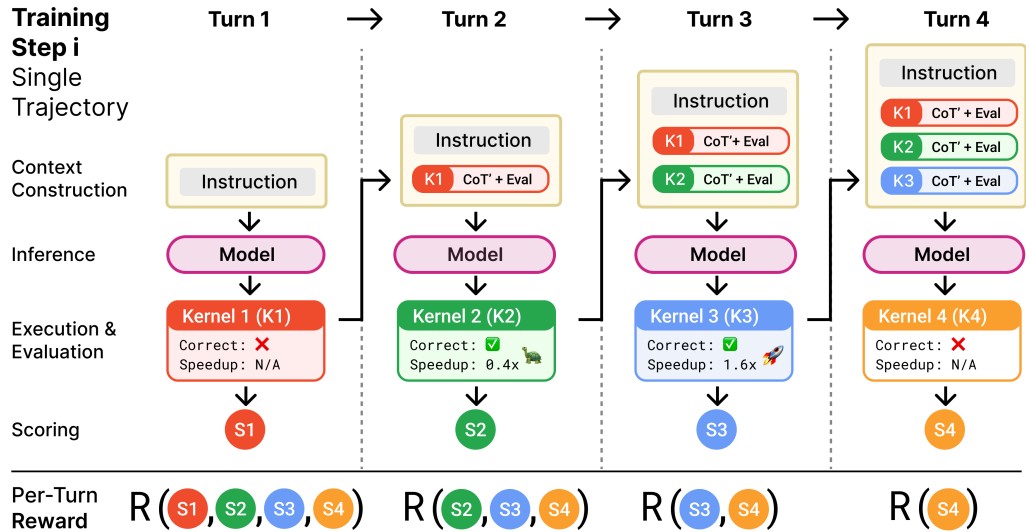

Figure 1: Within each training step, the model iteratively generates, executes, and refines kernels over multiple turns. Kernels are rewarded individually, based both on their performance and their contribution to subsequent speedups: `K1`, for example, while incorrect, leads to both a correct, slow kernel, `K2`, and a correct, performant kernel, `K3`, and should thus be rewarded accordingly. This setup enables Kevin to learn advanced code generation strategies that span multiple turns. Note: `CoT'` is the summarized chain of thought (CoT).

contrast, popular RL methods to train LLMs on verifiable rewards (Shao et al., 2024; Lambert et al., 2025) rely on the outcome reward of a single turn ("single-turn RL training"). We hypothesize that explicitly incorporating successive turns of code generation, execution, and feedback into each RL training step ("multi-turn RL training") better mirrors the iterative nature of kernel development, helping the model to learn more advanced code strategies that span multiple refinement turns.

We design a simple yet effective multi-turn RL training recipe, shown in Figure 1, that addresses the *key challenges* of training for CUDA kernel generation and optimization:

1. **Long trajectories lead to sparse rewards and context explosion**. To improve sample efficiency, we split trajectories and use each turn as an individual training sample. To address context explosion from long CoTs while preserving reasoning information, we summarize CoTs of prior turns.

2. **Finding an optimal solution may require rewarding suboptimal kernels that eventually lead to more performant ones.** Therefore, we study approaches to aggregate intermediate rewards across turns, finding a configuration that balances the correctness-performance trade-off.

3. **Reward hacking is prevalent as kernel generation is an open-ended, real-world engineering task:** e.g. the model can trick the evaluation harness, lazily copying the reference implementation instead of actually implementing kernels. To prevent this, we analyze the model's failure modes and enforce strict rule-based checks.

Enabled by our multi-turn RL training method on 180 KernelBench tasks from Level 1 and 2, we present K(ernel D)evin, the first RL-trained model to generate CUDA kernels. We compare Kevin and other models in our evaluation setting on a representative KernelBench eval set. Kevin improves upon its base model QwQ-32B, (Team, 2025d) both in correctness (56% → 82%) and mean speedup of generated kernels (in pure CUDA): from 0.53x to 1.10x over PyTorch Eager, while outperforming frontier models like OpenAI o4-mini (0.78x).

We then study the characteristics of Kevin in a test-time scaling setting, comparing it to a single-turn RL-baseline. We systematically compare the benefits of scaling along two axes of test-time compute: sequentially with more refinement turns (Ehrlich et al., 2025; Wang et al., 2025a) or in parallel with more trajectories (Brown et al., 2024; Snell et al., 2024). In our setting, we find that sequential scaling is much more effective, highlighting the importance of iterating upon execution feedback. We observe

that the model trained with multi-turn RL exhibits better scaling characteristics with more refinement turns, compared to the base model and the single-turn RL baseline. Our core contributions include:

1. **We design an effective yet flexible multi-turn RL training strategy that significantly improves model's capability on CUDA kernel generation**. This strategy addresses challenges that arise in real-world settings, and may be applicable to other environments that benefit from iterative optimizations.

2. **We found multi-turn is more effective both for training and inference** through systematic ablations: the multi-turn trained model outperforms the single-turn trained model across different evaluation setups. Furthermore, we found multi-turn inference is more effective across both models under a fixed inference budget.

3. **Kevin exhibits strong test-time scaling trends on both serial and parallel axes**, with a faster rate of improvement than its single-turn RL counterpart and its base model, while maintaining exploration capacity.

## 2 BACKGROUND AND RELATED WORK

### 2.1 LLM FOR GPU KERNEL OPTIMIZATION

There has been a surge of interest in exploring how to leverage LLMs to generate GPU kernels (NVIDIA, 2025), driven by the high cost and the long engineering cycles required to develop them (e.g. 2 years for efficient FlashAttention (Dao, 2023) port after Hopper GPU release). However, frontier models underperform on representative benchmarks like KernelBench (Ouyang et al., 2025) and TritonBench (Li et al., 2025), likely due to GPU code being severely underrepresented in the training data (CUDA, for example, accounts for less than 0.1% of pretraining data in the Stack (Kocetkov et al., 2022; Li et al., 2023)). Collecting more expert-written code is expensive, as only a limited number of developers are able to implement high-quality kernels. To tackle this task, there has been an explosion of agentic systems (Damani et al., 2024; Chen et al., 2025; METR, 2025) with custom workflows and evolutionary search methods (Lange et al., 2025; Google DeepMind, 2025). Yet these approaches typically incur high inference cost — e.g. $15 per kernel (Lange et al., 2025). Improving the base LLM's kernel-generation ability is therefore essential — and could significantly boost the efficiency for downstream agentic workflows.

### 2.2 RL OPTIMIZATION FOR LLMS TARGETING VERIFIABLE DOMAINS

Reinforcement Learning techniques like GRPO (Shao et al., 2024) have been shown to significantly enhance LLMs' performance on verifiable domains (Lambert et al., 2025) such as math (Team, 2025c; Wang et al., 2025b) and competitive programming (Team, 2025d; Luo et al., 2025a;b). These approaches can be further adapted for real-world software tasks, using fine-grain unit tests (Liu et al., 2023) or comparisons between code edits (Wei et al., 2025) as outcome rewards. Existing methods for code optimizations — where objective concerns performance beyond correctness — have been largely confined to supervised fine-tuning (Waghjale et al., 2024) and imitation learning (Shypula et al., 2024), highlighting Kevin's RL approach a novel contribution for this setting.

Given that tasks like performance optimization or long-horizon planning require multiple sequences of interrelated actions, several works (Goldie et al., 2025; Cao et al., 2025; Wang et al., 2025c; Zhou et al., 2024; Zhuang* et al., 2025) have explored RL training for multi-turn optimizations beyond optimizing for outcome from a single turn. Specific for the code setting, RLEF (Gehring et al., 2025) frames code generation as a multi-turn RL task: the model is allowed a fixed number of refinements turns and assigned a single binary pass/fail reward for final generation — training with such an approach might present sample-inefficiency issues. Unlike RLEF, which assigns rewards only at the final turn, our multi-turn RL framework for Kevin trains on every turn regardless of how optimal the code is, and optimizes for performance beyond just correctness. It is worth noting that Kevin's multi-turn RL training could be viewed as a variant of Meta-Learning (Xiang et al., 2025; Duan et al., 2016) or In-Context Reinforcement Learning (Nie et al., 2024; Tajwar et al., 2025; Schmied et al., 2025), where the focus is to improve solution quality during test-time with feedback (Qu et al., 2025); but adapted in a novel way to the challenging real-world setting of GPU kernel generation and code optimization.

## 3 TASK AND BASELINE

### 3.1 ENVIRONMENT AND EVALUATION

We use KernelBench (Ouyang et al., 2025), a popular dataset for evaluating the LLMs' ability to generate CUDA kernels for deep learning workloads in PyTorch. We chose 180 of both 100 Level 1 problems (basic operators: convolutions, matrix multiplies, loss functions, etc.) and 100 Level 2 problems (sequences of operators with fusion opportunities) as training environments. Since current KernelBench does not provide a train-test split, we construct 80 additional novel tasks following the same methodology (see Appendix A). We build the evaluation set by combining our 80 newly created tasks with the 20 remaining original KernelBench tasks, for a total of 100 evaluation tasks.

Each KernelBench task consists of generating a CUDA kernel given a PyTorch reference implementation, which is used to evaluate correctness and speedup. In our setup, we evaluate the model-generated kernels as follows: we verify the output is in the correct format (ensure resultant code is only implemented with inline CUDA) and check for reward hacking (Section 6.2). We then evaluate the kernel for compilation, runtime errors, and correctness. If the implementation is correct, we profile the kernel for its runtime.

### 3.2 KERNEL SCORE DESIGN

As we are concerned both with correctness and speedup, we assign a score $S$ for each kernel evaluation result that effectively balances the correctness-performance trade-off.

$$S = 0.3 \cdot \mathbf{1}_{\{\text{correct}\}} + \frac{T_{\text{baseline}}}{T_{\text{kernel}}} \cdot \mathbf{1}_{\{\text{correct}\}}$$

Correctness is checked against the reference program when tested with randomized inputs; speedup is computed as the ratio between PyTorch baseline time and kernel runtime. We experimented with various weights of correctness and speedup, finding this configuration through ablations on models ranging from 7B to 32B.

In addition, we explored rewarding intermediate objectives (successfully compile or execute), yet this caused model to over-optimize for intermediate steps (e.g. generating kernels that only compile, but are not necessarily correct). We also experimented with a length penalty on the response, as suggested by Team (2025b), but found that it degrades our model's performance during training.

### 3.3 SINGLE-TURN TRAINING

We apply GRPO (Shao et al., 2024) to train the model on kernel generation without iterating on external feedback ("single-turn" training). In each training step, we sample 16 responses per task and assign the evaluated score as the reward for each kernel. We compute the GRPO loss according to (Shao et al., 2024), which updates the policy by maximizing the following objective:

$$\mathcal{J}_{GRPO}(\theta) = \mathbb{E}[q \sim P(Q), \{o_i\}_{i=1}^{G} \sim \pi_{\theta_{\text{old}}}(O|q)]$$

$$\frac{1}{G} \sum_{i=1}^{G} \frac{1}{|o_i|} \sum_{t=1}^{|o_i|} \left\{ \min\left[ \frac{\pi_\theta(o_{i,t}|q, o_{i,<t})}{\pi_{\theta_{\text{old}}}(o_{i,t}|q, o_{i,<t})} \hat{A}_{i,t}, \text{clip}\left( \frac{\pi_\theta(o_{i,t}|q, o_{i,<t})}{\pi_{\theta_{\text{old}}}(o_{i,t}|q, o_{i,<t})}, 1 - \epsilon, 1 + \epsilon \right) \hat{A}_{i,t} \right] - \beta D_{KL}(\pi_\theta || \pi_{\text{ref}}) \right\}$$

$$(1)$$

where $\hat{A}_{i,t} = \frac{r_i - \text{mean}(\mathbf{r})}{\text{std}(\mathbf{r})}$, and $r_i$ is the score of a specific kernel.

We choose `Qwen QwQ-32B` (Team, 2025d) as base model. See Appendix B.6 for the rationale.

Following Yu et al. (2025), we apply `Clip-Higher`. We sample with `temperature` $= 0.9$ for both training and inference. We set the KL coefficient to $0$ to allow the model to deviate freely from the base policy, following Luo et al. (2025a).

We observe that reward plateaus after 100 gradient steps, likely because single-turn training prevents the model from refining its kernels. Many generated kernels are nearly correct–often a syntax or compilation fix away–but still receive 0 reward, discouraging the model from producing them. Similarly, the correct kernels do not achieve high speedup, as the model optimizes for correctness rather than attempting a risky approach. We address these limitations through multi-turn training.

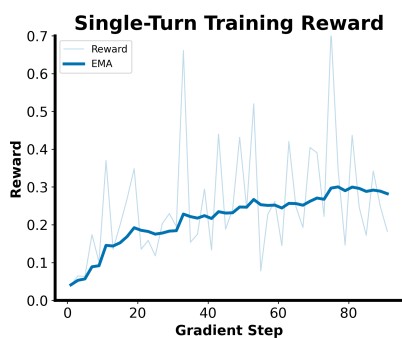

Figure 2: **Reward plateaus during single-turn training.**

## 4 MULTI-TURN TRAINING

In each multi-turn training step:

1. For each task, we sample m parallel trajectories of n sequential refinement turns each. Within each turn, the model generates: (1) chain-of-thought reasoning, (2) the kernel itself wrapped in a code block, and (3) a summary of the chain of thought (including the main implementation techniques used). We construct the training sample of that turn by including all of (1), (2), (3) in the response, over which we compute the loss. However, in subsequent turns, only (2) and (3) are kept in the context of the model.

2. We construct the context of a sample by including the history of previous responses, which include generated kernels along with their summarized CoTs, and evaluation feedback.

3. We evaluate the generated kernel and compute its score as shown in Section 3.2. The reward of each turn (CoT + response) is the discounted sum of current and subsequent scores, which we elaborate in Section 4.3.

4. For each task, we normalize the rewards across the $mn$ samples for advantage calculation. Then we compute the GRPO loss over the entire batch.

### 4.1 MANAGING CONTEXT

Reasoning models generate long CoTs, especially for complex tasks like kernel generation. Including all CoTs causes the context to grow rapidly, reaching 50-100k tokens within a few turns, surpassing the model's context length. To prevent context explosion, we discard CoTs of previous turns; yet to preserve information regarding the reasoning process, we ask the model to summarize the changes applied. This summary, along with the generated kernels and evaluation results, is passed to subsequent turns.

### 4.2 TRAINING ON EVERY REFINEMENT TURN

A reasonable approach for multi-turn RL is to treat an entire n-turn trajectory as a single training sample with the final outcome reward, as in RLEF (Gehring et al., 2025). However, this approach cannot attribute credit to individual turns within the trajectory, which we motivate could be important in the kernel setting in Figure 1. Instead, we split each n-turn trajectory into n individual training examples. Each training sample corresponds to a single turn within the trajectory and includes: (1) the complete CoT and kernel from that turn, and (2) the context history from previous turns (summarized CoT, kernels, and evaluation results). This turn-by-turn formulation enables direct reward assignment to each turn, substantially improving sample efficiency.

### 4.3 REWARD AGGREGATION AND DISCOUNTING

We initially explored two naive strategies for multi-turn credit assignment. The greedy approach assigns to each turn its corresponding kernel score, while the outcome-based approach assigns to all

turns the best score in the trajectory. The former failed to reward early suboptimal turns that lead to performant kernels later, while the latter ignores individual contributions and is sample inefficient.

Our method balances both approaches by aggregating the future kernels scores with a discount factor. We conduct ablations on the reward formulation. For score aggregation, we can either take the sum $R_t = \sum_{i=t}^{T} \gamma^{i-t} r_i$ or maximum $R_t = \max_{i=t,...,T} \left\{ \gamma^{i-t} r_i \right\}$ over future scores. Sum favors generating multiple good kernels, while max prioritizes achieving one high-performing kernel. We evaluate both forms with $\gamma = 0.4$ and $\gamma = 0.8$.

Experiments show that sum with $\gamma = 0.4$ scales best over 8 turns, though max performs better with $\gamma = 0.8$ with fewer turns. We decide to use the sum reward formulation with discount factor $\gamma = 0.4$.

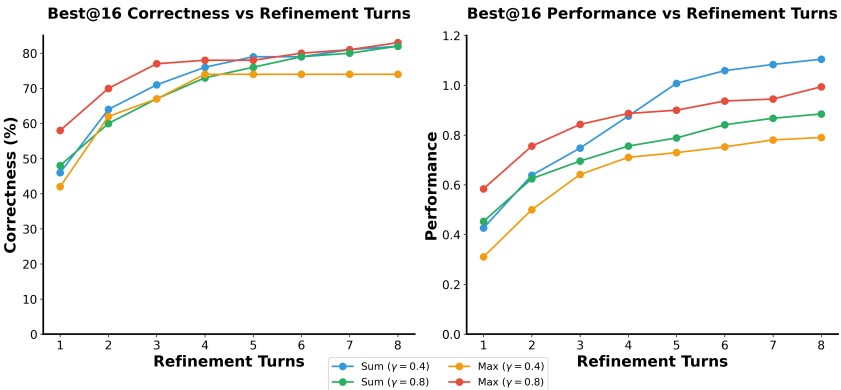

Figure 3: **Sum with $\gamma = 0.4$ exhibits the best scaling behavior.** We evaluate models trained with different reward formulations under 16 parallel trajectories and 8 refinement turns.

### 4.4 MULTI-TURN TRAINING BEHAVIOR

For multi-turn ablations and training runs, we train to 80 gradient steps; within each step, for each task, we sample 16 parallel trajectories and conduct 4 refinement turns. Each batch contains 8 tasks. (See Appendices B.5 for detailed hyperparameters and C.1 for training statistics)

Unlike single-turn training, reward now steadily increases. We also observe response-length behaviors similar to Luo et al. (2025b): the response length initially decreases, and then it starts increasing again as the model attempts more sophisticated solutions; we extend the max response length from 16K to 22K tokens at gradient step 60.

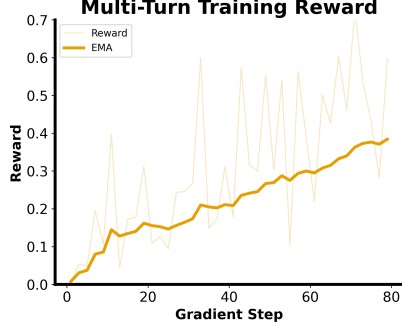

Figure 4: **Reward climbs steadily for multi-turn training.**

## 5 EVALUATION

As kernel generation is a challenging task, models are often given extensive test-time compute to tackle it. At inference, we employ multiple parallel trajectories, each made up of several serial turns.

We mark a trajectory **correct** if it contains at least one correct kernel. Its **performance** is the speedup of the fastest kernel (within the trajectory) over the PyTorch Eager reference (speedup of 0x if no kernel is correct). We also consider the **fast$_p$** metric, introduced by Ouyang et al. (2025), which is a binary indicator for whether a trajectory contains a correct kernel with performance of $p$ or more. To aggregate a metric across $k$ parallel trajectories for a given task, we compute: **best@k**, the maximum for that metric across all trajectories; **avg@k**, the average value across trajectories.

## 5.1 RESULT ON KERNELBENCH EVAL SET

We compare Kevin against frontier models and the single-turn RL baseline on our aforementioned KernelBench eval set of 100 tasks (Section 3.1), with 16 parallel trajectories, 8 serial refinement turns. As shown in Table 1, Kevin achieves a higher performance than its single-turn trained counterpart and other frontier models, demonstrating significant improvement from its base model (`QwQ-32B`). Qualitatively, Kevin is able to more effectively implement more aggressive optimizations across several turns (see Appendix H for examples); see Appendix E for additional evaluation details.

| Model | Correctness | | Performance | | $\text{fast}_1$ | | $\text{fast}_{1.5}$ | |
|---|---|---|---|---|---|---|---|---|
| | best@16 | avg@16 | best@16 | avg@16 | best@16 | avg@16 | best@16 | avg@16 |
| Kevin (Multi-Turn) | **82%** | **46%** | **1.10x** | **0.40x** | **43%** | 15% | **20%** | **6%** |
| Single-Turn RL | **82%** | 45% | 0.85x | 0.35x | **43%** | **16%** | 16% | 4% |
| Qwen QwQ-32B | 56% | 11% | 0.53x | 0.08x | 23% | 3% | 10% | 1% |
| OpenAI o4-mini | 38% | 22% | 0.78x | 0.27x | 21% | 7% | 13% | **6%** |
| OpenAI o3-mini | 27% | 8% | 0.30x | 0.08x | 9% | 2% | 4% | 2% |

Table 1: **Kevin (multi-turn RL) outperforms other models in correctness and performance.** We evaluate on 100 unseen KernelBench tasks with 16 parallel trajectories and 8 refinement turns.

## 5.2 SCALING REFINEMENT TURNS

Leveraging execution feedback is crucial at test time (Ehrlich et al., 2025; Wang et al., 2025a). Thus, we evaluate how Kevin scales with additional refinement turns. As shown in Figure 5, the single-turn model achieves slightly better performance with 1 turn, as its training objective optimizes for a single attempt. However, when given more refinement turns, the multi-turn trained model achieves significantly higher performance, with its curve showing the highest slope. This shows that multi-turn training enhances the model's ability to refine and optimize kernels over turns.

## 5.3 SCALING PARALLEL SAMPLES

We study how best@k performance scales when increasing the number of parallel trajectories $k$, while fixing the number of serial refinements turns. Prior work for RLVR on math problems (Yue et al., 2025) found that RL training limits models' exploration capacity, leading to worse best@$k$ metrics than the base model at large $k$. As shown in Figure 6, the performance curve of the single-turn RL model presents a lower slope compared to the base model, possibly hinting at this phenomenon. In contrast, our model trained with multi-turn RL achieves a higher slope compared to both the single-turn counterpart and the base model, suggesting that multi-turn training could maintain model's exploration capacity while improving model's performance.

## 5.4 PARALLEL VS SEQUENTIAL SCALING

As scaling test-time compute through parallel sampling (Snell et al., 2024) and sequential iterative refinement (Ehrlich et al., 2025) are both beneficial, we want to systematically compare their effectiveness for kernel generation. To investigate, we evaluate 3 inference-time configurations under the same total inference call budget (128 kernels): 128 trajectories with 1 turn, 32 trajectories with 4 turns, and 16 trajectories with 8 turns. As Table 2 shows, allocating more refinement turns during test-time is consistently better across various models, with 16 trajectories and 8 turns being optimal.

As Section 5.1 shows, multi-turn outperforms single-turn training when evaluated in a multi-turn inference setting. But since single-turn training optimizes for single-turn performance, a natural question arises: does the single-turn trained model perform better by generating more single-turn responses in parallel? In Table 2, we observe that in a single-turn inference setting with 128 parallel trajectories, the single-turn model achieves slightly better performance than the multi-turn model. However, when given more refinement turns at inference, the performance and correctness improve for all models. This strengthens the case for training a model that could use feedback effectively across

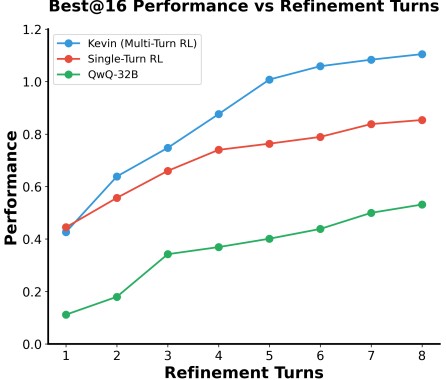

Figure 5: **Kevin effectively leverages multiple turns**. We evaluate the above checkpoints under the same environment with 16 parallel trajectories and 8 refinement turns.

Figure 6: **Multi-turn training maintains exploration capacity.** Refinement turns are fixed to 8, and best@k performance is computed with the estimator according to Chen et al. (2021).

multiple turns. Moreover, the multi-turn trained model achieves significantly higher performance, with faster improvement rates compared to the single-turn trained model at test-time.

| Model | Inference Config | | | Performance | Correctness |
|---|---|---|---|---|---|
| | Total | # Traj | # Turns | best@# traj | best@# traj |
| Multi-Turn RL | 128 | 16 | 8 | **1.10x** | 82.00% |
| Multi-Turn RL | 128 | 32 | 4 | 1.02x | **83.00%** |
| Multi-Turn RL | 128 | 128 | 1 | 0.65x | 76.00% |
| Single-Turn RL | 128 | 16 | 8 | **0.85x** | 82.00% |
| Single-Turn RL | 128 | 32 | 4 | 0.81x | 79.00% |
| Single-Turn RL | 128 | 128 | 1 | 0.70x | 73.00% |
| QwQ-32B | 128 | 16 | 8 | **0.53x** | **57.00%** |
| QwQ-32B | 128 | 32 | 4 | 0.47x | 52.00% |
| QwQ-32B | 128 | 128 | 1 | 0.42x | 54.00% |

Table 2: **Multi-turn inference with 16 trajectories and 8 turns is our most optimal setup,** when comparing inference configurations and their performance ($\times$ speedup) and correctness rates.

# 6 DISCUSSION

## 6.1 MODEL INSTABILITY

As prior RLVR work (Team et al., 2025) on `QwQ-32B` has shown, maintaining RL training stability is a recurring challenge. In our multi-turn setting, we notice distinctive patterns of instability, and develop a proxy signal that guides mitigation strategies. Specifically, we observe that training for longer often causes generation of repetitive and nonsensical outputs ("junk"). In the multi-turn case, junk first appears in the final turn and gradually spreads to earlier turns, leading to model collapse.

We identified a proxy signal, which we call the "Not Okay Ratio". `QwQ-32B` always

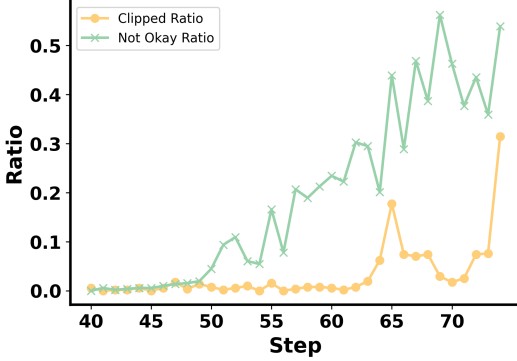

Figure 7: **"Not Okay Ratio" foresees model instability.** Here the proxy signal appears roughly 15 steps earlier than junk, which is indicated by the response "Clipping Ratio" metric (Luo et al., 2025b).

begins its chain of thought with `"Okay, "` but after 80 gradient steps, the model begins with erratic variants like `"Okay Amigos, so I need to optimize this 3D tensor-matrix multiplication"` and `"Okay Holy crap, I need to get this code optimized"`; tracking the "Not Okay Ratio" offers a reliable early proxy for model instability and well precedes junk.

As detailed in Appendix F, after attempting mitigations such as a KL penalty, we found that using constant-length normalization in the GRPO loss (Liu et al., 2025), together with gradient-norm clipping at 0.05, successfully delayed the onset of junk responses to beyond 100 gradient steps.

## 6.2 REWARD HACKING

We observe forms of reward hacking, as model capabilities fall short of task difficulty (Amodei et al., 2016). Concretely, when a weaker model such as `DeepSeek-R1-Distill-Qwen-7B` fails to produce the correct CUDA kernels, it resorts to directly copying the reference implementation, inheriting from it, or wrapping it in try-except statements. With a stronger prior like `QwQ-32B`, the model only fuses simple operators (ReLU, Max) and leaves key operators unmodified (in PyTorch). We address these issues by imposing stricter format checks that assign 0 reward to responses with *any* PyTorch functional operators. We elaborate on concrete examples in Appendix G.

## 6.3 DATA DISTRIBUTION

We found it critical to have a balanced difficulty distribution across the dataset, so that on average each batch contains both easier and harder tasks. In one experiment with `DeepSeek-R1-Distill-Qwen-14B` (DeepSeek-AI, 2025), we trained on a subset of only easy tasks. The reward quickly plateaus as the model overfits to a single difficulty level. Training with a stronger base model `QwQ-32B` and on both level 1 and 2 of KernelBench resolved the issue.

# 7 CONCLUSION

## 7.1 SUMMARY

We designed a multi-turn RL training recipe that addresses challenges when applied to the real-world task of kernel generation: specifically, effective context management and credit attribution across every turn to enable better sample efficiency. We also added safeguards against reward hacking, and experimented with approaches to constrain and predict instability.

We present Kevin, the first model trained with RL to generate CUDA kernels. Evaluated on an unseen evaluation set, Kevin outperforms both its single-turn RL counterpart and frontier models, demonstrating that our training recipe enables the model to learn more effective refinement strategies. Multi-turn training also enables better test-time scaling, both when increasing sequential refinement and parallel sampling compute, while preserving the exploration capacity of the model.

## 7.2 LIMITATIONS

Our work is limited by the number of robust tasks in kernel generation (unlike math or general coding with thousands of readily available tasks). KernelBench contains only 250 tasks and requires substantial pre-processing (Appendix A). Moreover, multi-turn RL is computationally expensive, even after extensive system optimization (Appendix C), as each rollout involves serial steps of reasoning inference, complex code generation, and careful kernel evaluation.

Nonetheless, we believe that showing significant performance gains in this domain, even under limited data and compute, highlights the effectiveness of our multi-turn training recipe. With more robust kernel environments, stronger model priors, and improved RL frameworks, we expect our method to scale accordingly.

We further note as KernelBench tasks are specified with pre-defined tensor input sizes, the speedups we measure in Section 3.2 are only accurate for those dimensions and on NVIDIA H200 GPUs.

## 7.3 FUTURE WORK

We see several directions for extending our method. Incorporating a learned value network and PPO (Schulman et al., 2017) may improve baseline estimation. More sophisticated search methods (beam search, MCTS (Silver et al., 2017)) may be applied at train and test time. Inspired by recent works (Sareen et al., 2025), the value network could also serve as a verifier for search at test-time.

Our multi-turn RL recipe demonstrates success in the real-world engineering task of GPU kernel generation. We hope our flexible design could be applicable to a wider range of tasks with verifiable rewards and execution feedback across a trajectory. We believe explicitly training models to reason about complex tasks over multiple turns is a key step towards enabling autonomous AI systems.

## 8 ETHICS STATEMENT

This work introduces Kevin, a multi-turn RL training method to enhance LLM's ability specifically for the task of automatic kernel generation. Our research builds on the publicly available model of `QwQ-32B` (Team, 2025d) and KernelBench dataset (Ouyang et al., 2025). We document in-depth how we use the dataset and post-train the model.

Our work does not introduce new risks that are not already inherent in the underlying base model. We do not involve any human subject nor do we make comparison with human kernel engineers in our study, as our baseline comparisons are against the PyTorch framework (Ansel et al., 2024), following the evaluation methodology proposed in KernelBench.

## 9 REPRODUCIBILITY STATEMENT

**Training Recipe**: We cover various challenges encountered during training in detail and propose effective mitigation: covering training stability F, avoiding reward hacking G, and careful considerations for RL design 4 with ablation studies. We elaborate on how we conduct dataset processing A) and provide a comprehensive set of hyper-parameters for our final model (AppendixB.5).

**Computational Requirements:** Each of our multi-turn training runs (for ablations and the final run) requires 650 H200 hours. As discussed in Appendix C, we take steps to improve the training efficiency of this complex multi-turn RL pipeline with in-the-loop kernel profiling. We elaborate on the computation cost and step time in Appendix C.1 and specifically in Table 4.

**Hardware Specifications:** We conduct all of our RL training, evaluation, and inference on the NVIDIA H200 platform. All of our kernel runtime measurement and baseline are specific to PyTorch 2.6 and H200 hardware.

**Model Weights:** Model weights will be released as open source and will be accessible to ensure reproducibility.

**Evaluation:** For our result, we compare our methods with other models that are either released open-source (`QwQ-32B`) or using a fixed version of the cloud API endpoints (`o4-mini-2025-04-16`, `o3-mini-2025-01-31`).

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

## A KERNELBENCH MODIFICATIONS

We use KernelBench Ouyang et al. (2025) as our training environments. KernelBench is a popular benchmark for evaluating LLMs' ability to generate performant CUDA kernels for deep learning workloads in PyTorch. Each KernelBench task consists in generating a CUDA kernel given a PyTorch reference implementation, which is used to evaluate correctness and speedup.

### A.1 TASK IMPROVEMENTS

We identify several limitations in the original KernelBench and introduce targeted modifications to address them. These changes are crucial to mitigate reward hacking, as shown in Section 6.2.

- We sand-boxed the kernel evaluation process so that fatal errors, such as `CUDA illegal memory accesses`, do not crash the RL training process.
- A significant issue we noted in KernelBench was that for many tasks, the input tensors used to measure performance are quite small. This causes kernel launch overhead to take up a significant portion of the runtime. To address this, we enlarged the tensor dimensions of the affected tasks.
- A sneakier bug in the KernelBench's evaluation harness caused the tested kernel to recycle the output tensor from the reference implementation (which was run immediately before) as its own tensor output. As a result of this, a kernel that only computes (correctly) a portion of the output tensor would still pass the correctness check. We address this by running the tested kernel first and only after the reference implementation, thus avoiding this hack.

In the end, we chose a total of 180 tasks as training environments, with 90 of the 100 Level 1 problems and 90 Level 2 problems (sequences of operators with fusion opportunities).

### A.2 CONSTRUCTION OF ADDITIONAL EVALUATION SET

Since current KernelBench does not provide a train-test split, we construct 80 additional tasks following the same methodology that KernelBench was constructed.

KernelBench Level 2 is constructed by composing a subset of PyTorch operators as sequences of operators. Specifically, the PyTorch operators are categorized as:

- **Main operators:** `Conv2d`, `Matmul`, `Gemm`, `BMM`, `Conv3d`, `ConvTranspose2d`, `ConvTranspose3d`.
- **Activations:** `ReLU`, `Sigmoid`, `Tanh`, `LeakyReLU`, `GELU`, `Swish`, `Softmax`, `Mish`, `Hardtanh`, `HardSwish`.
- **Element-wise operators:** `Add`, `Multiply`, `Subtract`, `Divide`, `Clamp`, `Scale`, `ResidualAdd`.
- **Normalizations:** `BatchNorm`, `LayerNorm`, `InstanceNorm`, `GroupNorm`.
- **Pooling:** `MaxPool`, `AvgPool`, `GlobalAvgPool`.
- **Bias:** `BiasAdd`.
- **Reductions:** `Sum`, `Mean`, `Max`, `Min`, `LogSumExp`.
- **Others:** `ResidualAdd`, `Scaling`.

To construct the additional eval set (unseen from train set), following the methodology from original KernelBench task construction:

1. We sample from the available operators listed above: 1 main operator (computationally expensive), and 2-5 other operators.
2. We ask a language model, namely Gemini 2.5-Flash (Doshi, 2025), to generate a PyTorch program that creates a kernel by combining these operators. We also ask it to generate sample tensor sizes for the task.

3. We ensure this PyTorch program can be executed and has a runtime on NVIDIA H200 $> 0.1$ms, to avoid the runtime being dominated by kernel launch (CPU) overhead.

4. We make sure this PyTorch program (with the same sequence of operators) is not present in existing KernelBench Level 1 and 2 programs.

We manually inspected all new task programs to ensure their validity. We build the evaluation set by combining our 80 newly created tasks with the 20 remaining original KernelBench tasks, for a total of 100 unseen evaluation tasks.

## B    ADDITIONAL DETAILS ON MULTI-TURN RL

Here we elaborate on design choices for our RL Training as described in Section 3.3 and Section 4, along with some ablation results.

### B.1    MOTIVATION FOR TURN-WISE REWARD

In our multi-turn RL training setup, within each training step we have a trajectory with $n$ refinement turns. A possible approach would be to compute the reward based on the kernel at the last turn, similar to what is used in RLEF (Gehring et al., 2025). However, for the GPU kernel optimization setting, using just the last kernel might not be optimal at times: for example, as shown earlier in Figure 1, kernel 3 is correct but kernel 4 is incorrect as the model attempts more aggressive optimizations.

In this setting, computing reward based on the best kernel among the trajectory instead (max speedup) is a more natural choice. However, using only the max kernel score forces us to discard all turns in a trajectory after the max turn, possibly wasting a significant amount of inference rollouts: In the previous example, we would have to completely discard the reasoning trace, code, and evaluation for kernel 4. Thus, we arrived at our approach in Section 4.3, which uses a discounted look-ahead max or sum, enabling more sample-efficient training.

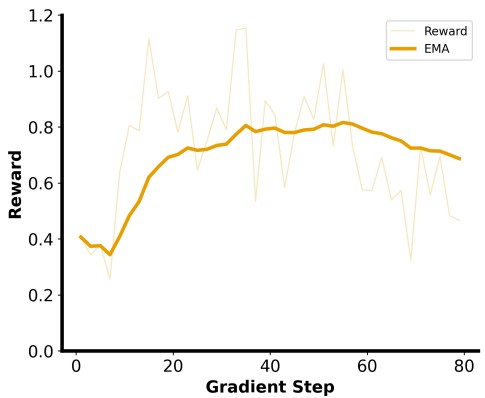 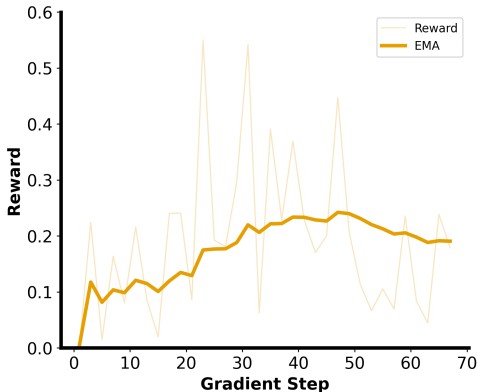

Figure 8: Training reward with correctness weighting of 1, performance / speedup weighting of 1. Concretely, $S = \mathbf{1}_{\{\text{correct}\}} + \frac{T_{\text{baseline}}}{T_{\text{kernel}}} \cdot \mathbf{1}_{\{\text{correct}\}}$.

Figure 9: Training reward with no correctness weighting, performance / speedup weighting of 1. (speedup is 0 if kernel is incorrect). Concretely, $S = \mathbf{1}_{\{\text{correct}\}} \cdot \frac{T_{\text{baseline}}}{T_{\text{kernel}}}$.

### B.2    WEIGHTING FOR SCORE

In Section 3.2, we explain our score design, which assigns a scalar value (score $S$) based on a kernel's correctness and speedup. We explore score design and how to balance the correctness-performance trade-off, after series of small-scale ablations on QwQ-32B (Team, 2025d).

We decided on a weighting of 0.3 on correctness and using speedup for performance (raw speedup itself, no weighting), which is $S = 0.3 \cdot \mathbf{1}_{\{\text{correct}\}} + \mathbf{1}_{\{\text{correct}\}} \cdot \frac{T_{\text{baseline}}}{T_{\text{kernel}}}$.

Here we present some ablation studies we ran with different weighting configurations for score design, particularly focusing on adjusting the weighing for correctness, in the context of single-turn RL (GRPO) training (as shown in Section 3.3). As show an example in Figure 8, where we set the weighting to 1.0 for correctness, the reward plateaus and eventually decreased; concretely, we observed that the model over-optimizes for generating correct kernels and does not explore speedup as much, causing the reward to plateau during training. In another experiment in Figure 9, we set the weighting to 0 for correctness, only rewarding the model for generating performant (and correct) kernels. We again observed the reward plateau. Thus, we hypothesize that it is still important to reward the model for correct kernels, as long as the correctness reward is not too significant, balancing the correctness-performance tradeoff.

### B.3 NUMBER OF TRAJECTORIES DURING TRAINING

We vary the number of parallel trajectories during Multi-Turn RL training (Section 4), using 64 parallel trajectories instead of 16 for each task during each training step. We note that best@16 correctness slightly increases, but the overall performance does not show significant improvements. Due to the high-compute requirements of doing more generations during training, we chose to train with 16 parallel trajectories.

### B.4 LENGTH PENALTY

We explore incorporating response length as a part of the reward design to incentivize the model to use its reasoning tokens more efficiently. We attempted a run using the length penalty from Kimi Team (2025b) on `DeepSeek-R1-Distill-Qwen-14B`. As shown in Figures 10 and 11, we found that the response length of the responses collapses, with the model no longer outputting CoT after 10 training steps, suggesting that the addition of a length penalty is counterproductive for our setting.

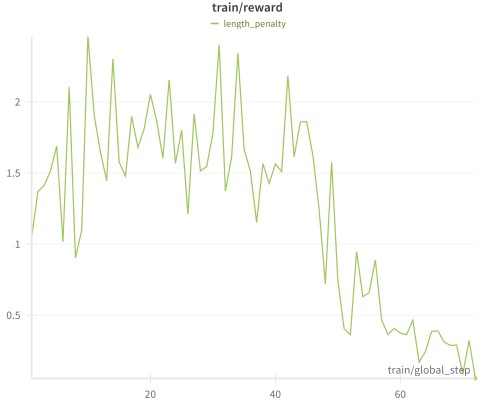

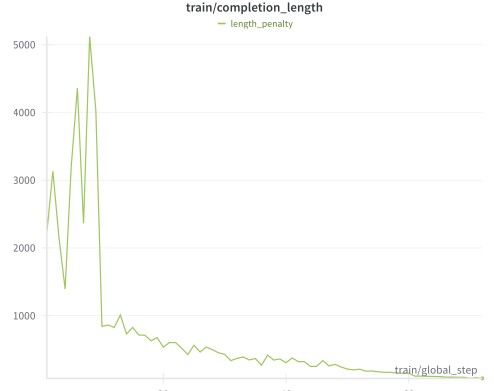

Figure 10: Training Reward collapses when including length penalty as part of reward

Figure 11: Response length of generations collapses when including length penalty as part of reward.

### B.5 DETAILED TRAINING HYPERPARAMETERS

Here is the set of hyperparameter for our final Kevin training run.

```
Constant learning rate of 2e-6 with warmup ratio of 0.03
Max grad norm = 0.05
KL coeff = 0
Temperature = 0.9
Top p = 0.95
Eps clip = 0.2 with clip high = 0.28
Max prompt len = 8192
Max generate len = 22432
```

### B.6 BASE MODEL CHOICE RATIONALE

We experimented with several different base models, such as `DeepSeek-R1-Distill-Qwen7B` DeepSeek-AI (2025), `DeepSeek-R1-Distill-Llama8B`, and Gemma27B-Instruct Team (2025a). These models, however exhibit weak kernel writing priors, which causes the initial reward to be overly sparse for effective learning, making the model resort to reward hacking (see Section 6.2).

Specifically, QwQ-32B exhibits stronger priors in math and coding than other base models, as shown in the the table below. Given that gradient updates only occur if the model receives non-zero reward in at least one rollout for a given task, and that for many tasks the model is unable to generate any correct solutions, the model only receives a very sparse signal for the few kernels that it's able to implement correctly. This sparse feedback is not sufficient for the model to learn. As a result, even after 15 steps, the reward shows no improvements. This behavior is reflected in the attempted training runs with these models: with the same recipe, there is no improvement in performance even after dozens of steps, with the reward staying flat.

| Model | AIME 24 | LiveCodeBench | SWE-Bench Verified | Aider Polyglot |
|---|---|---|---|---|
| QwQ-32B | **79.5%** | **63.4%** | **41.3%** | **20.9%** |
| Gemma27B-Instruct | 25.3% | 29.7% | N/A | 4.9% |
| DeepSeek-R1-Distill-Llama8B | 50.4% | 39.6% | N/A | N/A |
| DeepSeek-R1-Distill-Qwen7B | 55.5% | 37.6% | N/A | N/A |

Table 3: Base Model Comparison on Math and Coding Benchmarks

Here we also present the training reward curve for Gemma-27B. The model is unable to improve on the task since it receives little reward signal throughout the training run.

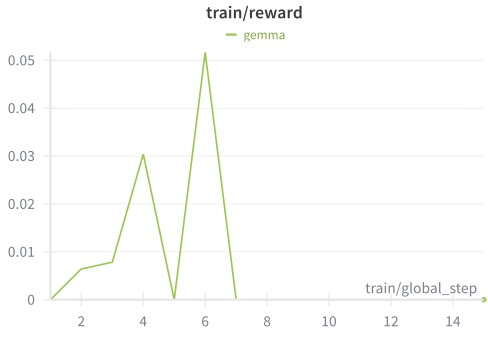

Figure 12: Training reward curve on Gemma-27B. With sparse reward signal which often leads to zero gradients, preventing model from improving on this task.

We thus choose `Qwen QwQ-32B` (Team, 2025d) as our base model, which exhibits, among all the models we have evaluated of comparable size, the strongest priors.

## C  RL INFRASTRUCTURE

Conducting RL training on a highly challenging task like GPU kernel generation is a computationally expensive process, requiring full-policy updates on a sufficiently capable base model, as discussed in Section 6.2.

Although a few open-source RL frameworks existed when we began this study, it is still difficult to support training in a kernel evaluation environment and including multiple turns within one training step. We built our training framework on top of the OpenRLHF (Hu et al., 2024) framework.

We use vLLM (Kwon et al., 2023) for inference and DeepSpeed Zero-3 (Wang et al., 2023) for offloading optimizer states.

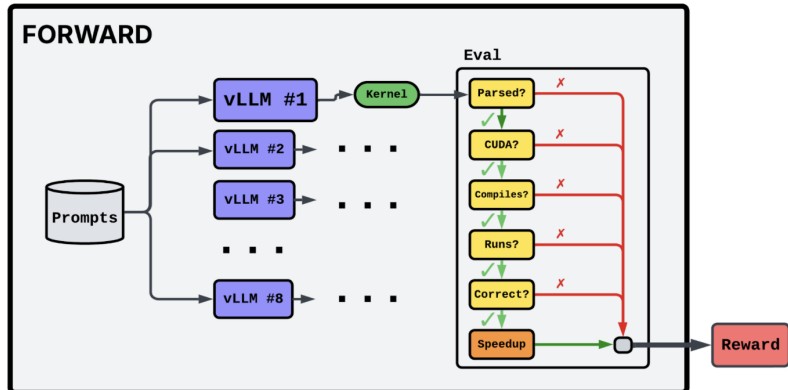

Figure 13: Overview of our RL Training infrastructure.

Each of the 8 GPUs handles the kernel generation and evaluation for one task. After the response generation finishes, each GPU offloads its vLLM engine to CPU memory and evaluates the kernels it generated. We run the evaluation and calculate reward and evaluation info. Each GPU then wakes up its corresponding vLLM engine and regenerates kernels.

We optimized our training infrastructure to co-locate vLLM rollout engines, kernel execution environments, and DeepSpeed trainers on the same GPU device, so other small research teams with 1 cluster could also experiment with our proposed method.

### C.1 TRAIN TIME STATISTICS

Here we elaborate more on the cost of our multi-turn training. The nature of multi-turn RL requires multiple serial turns of parallel rollouts and kernel compilation/execution after each step, making the overall training process compute-intensive. To accurately measure kernel runtime, we must clear the GPUs of any running processes and perform additional operations, such as warmup steps before profiling, which further limits the training speed. Here we show key training time statistics:

| Configuration | Value |
|---|---|
| Gradient steps | 80 |
| Parallel trajectory rollouts | 16 |
| Refinement turns (serial env. interactions) | 4 |
| Gradient updates per batch | 2 (1 on-policy, 1 off-policy) |
| Time for rollout + kernel execution (per step) | ~1.5 hours |
| Time for 1 gradient update (2 steps) | ~0.5 hours |
| Base model | QwQ-32B |

Table 4: Setup and Cost of multi-turn Training for Kevin on 8xH200s.

Overall, one training run (Section 4) takes $\tilde{6}50$ H200 hours, equivalent to around 3 days and 9 hours on a single node of 8xH200s. However, we believe concurrent and future systems projects (such as SkyRL by Cao et al. (2025)) will improve training efficiency, especially for roll-outs with complex interactions with environments. The demanding computational requirement of multi-turn RL is what leads us to focus on improving the sample efficiency of our method; specifically, we choose to train on every sample regardless of their performance and attribute credit effectively with our reward design.

## D INFERENCE SETUP

Our prompt is similar to the prompt used in KernelBench (Ouyang et al., 2025). We use this during training and test-time inference. In the first refinement turn, we add an example of the inline CUDA format to the prompt but remove it afterwards.

Below we show how we construct the context in the simplest case (of one turn, or the base prompt). In the context, we present model the KernelBench task, instructions, and a simple 1-shot example of a CUDA add kernel (to inform model the desired format for response):

```
You are given the following architecture:
import torch
import torch.nn as nn

class Model(nn.Module):
    """
    Simple model that performs Layer Normalization.
    """
    def __init__(self, normalized_shape: tuple):
        """
        Initializes the LayerNorm layer.

        Args:
            normalized_shape (tuple): Shape of the input tensor to be
    normalized.
        """
        super(Model, self).__init__()
        self.ln = nn.LayerNorm(normalized_shape=normalized_shape)

    def forward(self, x: torch.Tensor) -> torch.Tensor:
        """
        Applies Layer Normalization to the input tensor.

        Args:
            x (torch.Tensor): Input tensor of shape (*,
    normalized_shape).

        Returns:
            torch.Tensor: Output tensor with Layer Normalization
    applied, same shape as input.
        """
        return self.ln(x)

Replace pytorch operators in the given architecture with raw CUDA
    kernels, optimizing for performance on NVIDIA H200 (e.g. shared
    memory, kernel fusion, warp primitives, vectorization,...). Use
    torch.utils.cpp_extension.load_inline and name your optimized output
    architecture ModelNew. You are not allowed to use torch.nn (except
    for Parameter, containers, and init). The input and output have to
    be on CUDA device. Your answer must be the complete new architecture
    (no testing code, no other code): it will be evaluated and you will
    be given feedback on its correctness and speedup so you can keep
    iterating, trying to maximize the speedup. After your answer,
    summarize your changes in a few sentences.Here is an example:

import torch.nn as nn
from torch.utils.cpp_extension import load_inline

# Define the custom CUDA kernel for element-wise addition
elementwise_add_source = """
#include <torch/extension.h>
#include <cuda_runtime.h>

__global__ void elementwise_add_kernel(const float* a, const float* b,
    float* out, int size) {
    int idx = blockIdx.x * blockDim.x + threadIdx.x;
    if (idx < size) {
        out[idx] = a[idx] + b[idx];
    }
}
```

```
48  torch::Tensor elementwise_add_cuda(torch::Tensor a, torch::Tensor b) {
49      auto size = a.numel();
50      auto out = torch::zeros_like(a);
51
52      const int block_size = 256;
53      const int num_blocks = (size + block_size - 1) / block_size;
54
55      elementwise_add_kernel<<<num_blocks,
        block_size>>>(a.data_ptr<float>(), b.data_ptr<float>(),
        out.data_ptr<float>(), size);
56
57      return out;
58  }
59  """
60
61  elementwise_add_cpp_source = (
62      "torch::Tensor elementwise_add_cuda(torch::Tensor a, torch::Tensor
        b);"
63  )
64
65  # Compile the inline CUDA code for element-wise addition
66  elementwise_add = load_inline(
67      name="elementwise_add",
68      cpp_sources=elementwise_add_cpp_source,
69      cuda_sources=elementwise_add_source,
70      functions=["elementwise_add_cuda"],
71      verbose=True,
72      extra_cflags=[""],
73      extra_ldflags=[""],
74  )
75
76
77  class ModelNew(nn.Module):
78      def __init__(self) -> None:
79          super().__init__()
80          self.elementwise_add = elementwise_add
81
82      def forward(self, a, b):
83          return self.elementwise_add.elementwise_add_cuda(a, b)
```

For our multi-turn RL training (Section 4) and inference (Section 5), we provide model with the kernels, CoTs (summarized), and evaluation results of all previous turns in chronological order. We truncate the turns that do not fit inside the context window, starting from the earliest ones.

```
1   <Base prompt containing pytorch architecture and instruction>
2
3   Here are your previous attempts:
4
5   < for each (i) previously generated kernel >
6       <Previously generated kernel G[i]>
7
8       <Summary of CoT[i]>
9
10      <if parsing error>
11
12          Your previous answer failed to be parsed due to not adhering to
        the desired formatting. Here is the error message: <error_message>
13
14      <elif compilation error>
15
16          Your previous answer failed to compile. Here is the error
        message: <error_message>
17
18      <elif run error>
```

```
19
20        Your previous answer compiled successfully but had runtime
      errors. Here is the error message: <error_message>
21
22    <elif correctness error>
23
24        Your previous answer was incorrect. Here is the error message:
      <error_message>
25
26    <elif correct>
27
28        Your previous answer was correct but can be made faster. Here is
      the speedup you achieved relative to the baseline: <speedup>
29
30  Restart your reasoning process and generate new, complete code.
```

# E  ADDITIONAL EVALUATIONS

Here we present some additional evaluation results for Section 5.

## E.1  CONFIDENCE INTERVALS

We compute the confidence intervals of best@16 and avg@16 performance for the multi-turn and single-turn RL across 5 runs, as shown in Table 5. These results show multi-turn RL has statistically significant improvement on both metrics and hence its effectiveness.

| Model | Performance | |
|---|---|---|
| | best@16 | avg@16 |
| Multi-turn RL | $1.10 \pm 0.099$ | $0.40 \pm 0.011$ |
| Single-turn RL | $0.85 \pm 0.048$ | $0.35 \pm 0.013$ |

Table 5: **Evaluation on our evaluation set across 5 runs with confidence interval.** Multi-turn RL outperforms Single-turn RL on both best@16 and avg@16 performance.

## E.2  CHOICE OF BASELINE MODEL COMPARISON

Here we elaborate on the choice of model comparisons used for 5.1, notably against both Kevin's base model (QwQ-32B) and frontier reasoning models (o4-mini, o3-mini). To the best of our knowledge, we are not aware of any model specifically "fine-tuned" for the CUDA context (efforts like Nichols et al. (2024) focus on OpenMP CPU code). CUDA, or GPU code in general, is extremely sparse in the pretraining corpus, only 0.073% of the Stack (Li et al., 2023) code corpus; this makes approaches that depend on readily available data (such as "fine-tuning") difficult. Hence, this data challenge actually highlights the value of our RL-based approach, as we discussed in Section 1. We believe that the comparisons of Kevin against SoTA general-purpose LLMs are fair and fitting, and actually demonstrate the advantage of our RL-based approach in this domain.

Our baseline comparisons, o4-mini and o3-mini, are frontier models that achieve SoTA on challenging code generation benchmarks. Specifically we use o4-mini-2025-04-16 and o3-mini-2025-01-31. As shown below, o4-mini demonstrates a significant lead over our base model QwQ-32B, especially on challenging real-world software tasks such as SWE-Bench (Jimenez et al., 2024) and Polyglot (Gauthier, 2024). Hence, our results in Section 5.1 and Table 1 showing Kevin (post-trained QwQ-32B with multi-turn RL) exceeding o4-mini should be noted as a significant improvement and demonstrate our method's effectiveness.

| Model | AIME 24 | LiveCodeBench | SWE-Bench Verified | Aider Polyglot |
|-------|---------|---------------|--------------------|----------------|
| QwQ-32B | 79.5% | 63.4% | 41.3% | 20.9% |
| o4-mini | **93.4%** | **74.2%** | **68.1%** | **72.0%** |

Table 6: o4-mini shows significant lead over QwQ-32B over a variety of reasoning, coding, and software engineering benchmarks (Mathematical Association of America, 2024; Jain et al., 2024; Jimenez et al., 2024; Gauthier, 2024); Kevin is post-trained on QwQ-32B and shows improvement over both QwQ-32B and o4-mini, as shown in Section 5.1.

.

### E.3 EVALUATION ON KERNELBENCH LEVEL 3

While we focus on our training and evaluation mostly on KernelBench Level 1 and 2 (Section 3), we were also curious and explore testing Kevin on KernelBench Level 3 tasks. They are longer and more challenging (rather than single or a few operators), requiring the end-to-end optimization of full model architectures, such as the VisionTransformer, and miniGPT attention blocks. Kevin is trained using a subset (180) of the KernelBench Level 1 and 2 tasks (single and sequence of operators), and Level 3 tasks are completely unseen. We evaluate the multi-turn (Kevin), single-turn, and base model (QwQ-32B) on the 50 level 3 tasks following the same evaluation setup as Section 5. As shown in the table below, multi-turn RL can also generate much faster kernels for these much more complex tasks over both single-turn RL and the base model.

We view Level 3 primarily as an out-of-distribution test: these tasks involve full model architectures with much longer-horizon reasoning, and requiring both kernel generation and effectively dealing with long context. We do not train on any Level 3 tasks as the length of these programs would lead to context explosion (Section 4.1). Hence, our main analysis focuses on Levels 1 and 2, which better focuses on kernel generation performance with more controlled conditions.

| | **Correctness** | | **Performance** | |
|--------|---------|--------|---------|--------|
| **Method** | **best@16** | **avg@16** | **best@16** | **avg@16** |
| Multi-turn RL | **36%** | **11.75%** | **0.41** | **0.08** |
| Single-turn RL | **36%** | 8.38% | 0.36 | 0.06 |
| QwQ-32B | 4% | 0.25% | 0.04 | 0.002 |

Table 7: Multi-turn RL achieves improvements also on the completely unseen and more complex KernelBench Level 3.

### E.4 EVALUATION AGAINST TORCH.COMPILE BASELINE

torch.compile uses various rules and heuristics to optimize the performance of a PyTorch program, so in theory it is a stronger baseline. However, it is a less reliable baseline than PyTorch eager. As pointed out in KernelBench ((Ouyang et al., 2025)), the runtime can vary significantly across hardware platforms). Nevertheless, to substantiate the validity of our results against a potentially stronger baseline, we run additional evaluation here with the baseline computed by torch.compile default Inductor backend.

Under both the eager baseline and the torch compile baseline, multi-turn's best@16 performance exceeds both its single turn counterpart as well as all other models evaluated. We note that torch compile is effective when the computation graph is composed of a large number of operators. Given that we focus on optimizing the performance of either a single or a few operators, the impact of torch compile tends to be small, and for some kernels even negative, given the additional runtime launch overhead that torch compile introduces.

| Method | Correctness | | Speedup (eager) | | Speedup (compile) | |
|---|---|---|---|---|---|---|
| | best@16 | avg@16 | best@16 | avg@16 | best@16 | avg@16 |
| Multi-Turn RL | **82%** | **46%** | **1.10x** | **0.40x** | **1.04x** | 0.36x |
| Single-Turn RL | 82% | 45% | 0.85x | 0.35x | 0.91x | **0.37x** |
| QwQ-32B | 56% | 11% | 0.53x | 0.08x | 0.51x | 0.08x |
| sonnet-4 | 71% | 34% | 0.69x | 0.26x | 0.71x | 0.27x |
| o4-mini | 38% | 22% | 0.78x | 0.27x | 0.61x | 0.20x |
| gpt-4.1 | 36% | 14% | 0.25x | 0.09x | 0.24x | 0.09x |
| o3-mini | 27% | 8% | 0.30x | 0.08x | 0.22x | 0.07x |

Table 8: Correctness and speedup comparison across models.

### E.5 EVALUATION ON A100

In 7.2, we mentioned that speedups are only accurate for the pre-defined tensor dimensions and on NVIDIA H200 GPUs. To demonstrate the effectiveness of multi-turn RL transfers to other hardware architecture, we use the same evaluation setup on NVIDIA A100 GPUs, which use the Ampere architecture (instead of Hopper for H200s).

It is important to note that the models we evaluate were trained with execution feedback on H200s, but the speedup and correctness they see at inference-time are based on A100s, where the hardware layout is different. For example, Hopper GPUs have up to 228 KB of shared memory per SM, but Ampere GPUs have only 164 KB, so the trained models might think they have access to more shared memory than they actually do, leading to errors in the kernel refinement process. In spite of the difference in the hardware architecture during evaluation vs training, our results show the same trend as H200s, that multi-turn RL achieves better best@16 and avg@16 correctness and speedup than single-turn RL and the base model.

| Method | Correctness | | Performance | |
|---|---|---|---|---|
| | best@16 | avg@16 | best@16 | avg@16 |
| Kevin-32B | **79%** | **44%** | **0.88** | **0.33** |
| Single Turn | 78% | 42% | 0.71 | 0.29 |
| QwQ-32B | 53% | 12% | 0.45 | 0.08 |

Table 9: Multi-turn RL generalizes to other hardware architectures, also surpassing the single-turn RL and base models.

## F TRAINING STABILITY

The analysis of the "not okay ratio" led us to believe that model instability caused the appearance of nonsensical and repetitive outputs. Therefore, we attempted runs where we enabled KL divergence penalty in the GRPO loss, which would penalize the model from deviating from the base policy too much. Following DeepScaleR (Luo et al., 2025b), we set the KL coefficient to 0.001 and attempted an ablation run. However, we found that the reward plateaus with KL enabled, suggesting that the KL penalty slows down learning. Thus we attempted other techniques of constraining the model from deviating into regions of instability, such as clipping the gradient norm aggressively — which was effective in our setting.

We use 4 refinement turns at train-time for efficient training. During test time, we can afford more extensive test-time compute, so we evaluate on 8 turns instead of 4 turns.

## G REWARD HACKING

Here we present excerpts from generated kernels that show signs of reward hacking, previously mentioned in Section 6.2.

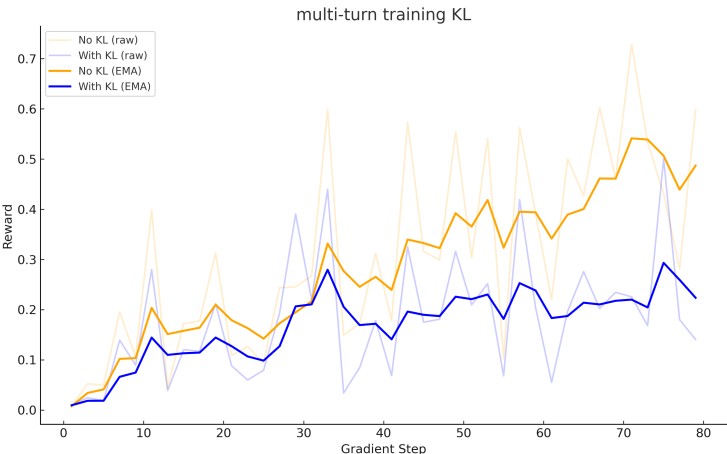

Figure 14: **Adding a KL penalty slows down learning.** Here we conduct an ablation with KL coefficient $\beta = 0.001$ versus $\beta = 0$. We see that the reward plateaus with KL enabled.

In the following example, the model simply copies the PyTorch reference implementation, thus getting rewarded for generating a correct answer with 1.0x speedup. To prevent this, we modify our kernel evaluation environment so that it checks each generated kernel if it contains instances of `torch.nn` or `torch.nn.functional`. We assign a reward of 0 to those.

```
class ModelReLU(Module):
    ...
    def forward(self, x):
        relu = torch.nn.ReLU()
        return relu(x)
```

Similarly, the model wraps an incorrect implementation of the CUDA kernel in a try-except statement and invokes the PyTorch implementation functions as a fallback. To prevent this, we assign a reward of 0 to kernels that contain try or except.

```
class ModelReLU(Module):
    ...
    def forward(self, x):
        try:
            ...   \# CUDA implementation
        except Exception as e:
            print("Custom ReLU kernel failed to compile. Using default
    ReLU instead.")
            relu = torch.nn.ReLU()
            return relu(x)
```

Here the model inherits from the reference implementation, bypassing the need for a CUDA implementation. To prevent this, we assign a reward of 0 to kernels that contain pass.

```
class ModelReLU(Module):
    ...
    def forward(self, x):
        ...   \# CUDA implementation

class ModelReLU(Model):  \# Model is the the reference implementation
    pass
```

## H  KERNEL HIGHLIGHT

As an example, we look at one the task form our evaluation set, `Conv3D_GELU_Tanh_Clamp`:

```python
1  import torch
2  import torch.nn as nn
3
4  class Model(nn.Module):
5      """
6      Model using Conv3d, GELU, Clamp, Tanh
7      """
8      def __init__(self, in_channels, out_channels, kernel_size,
       clamp_min, clamp_max):
9          super(Model, self).__init__()
10         self.conv = nn.Conv3d(in_channels, out_channels, kernel_size)
11         self.clamp_min = clamp_min
12         self.clamp_max = clamp_max
13
14     def forward(self, x):
15         """
16         Forward pass of the model.
17
18         Args:
19             x (torch.Tensor): Input tensor of shape (batch_size,
       in_channels, depth, height, width).
20
21         Returns:
22             torch.Tensor: Output tensor of shape (batch_size,
       out_channels, depth-kernel_size+1, height-kernel_size+1,
       width-kernel_size+1).
23         """
24         x = self.conv(x)
25         x = torch.tanh(x)
26         x = torch.clamp(x, min=self.clamp_min, max=self.clamp_max)
27         x = torch.nn.functional.gelu(x)
28         return x
```

The task is made up of a computationally intensive function (`3D convolution`) and a series of lightweight functions (`tanh`, `clamp`, `gelu`).

Note: all the speedups reported below are measured over torch in eager mode, as 'torch.compile''s support varies heavily across different platforms. When compiled using torch.compile, the kernel achieves a speedup of 1.32x over its eager counterpart. Thus, this trajectory achieves a maximum speedup of 1.93x over eager and 1.46x over torch compile.

The model implements a correct solution in the first turn by fusing the kernels, but it only achieves a speedup of **1.06x** over the reference implementation.

```python
1  import torch
2  from torch.utils.cpp_extension import load_inline
3  import math
4
5  class ModelNew(torch.nn.Module):
6      def __init__(self, in_channels, out_channels, kernel_size,
       clamp_min, clamp_max):
7          super(ModelNew, self).__init__()
8          self.clamp_min = clamp_min
9          self.clamp_max = clamp_max
10
11         # Initialize convolution weights and bias
12         self.weight = torch.nn.Parameter(torch.empty(out_channels,
       in_channels, kernel_size, kernel_size, kernel_size))
13         self.bias = torch.nn.Parameter(torch.empty(out_channels))
14         # Weight initialization following PyTorch's Conv3d default
15         torch.nn.init.kaiming_uniform_(self.weight, a=math.sqrt(5))
16         fan_in, _ =
       torch.nn.init._calculate_fan_in_and_fan_out(self.weight)
17         bound = 1 / math.sqrt(fan_in)
```

```
18              torch.nn.init.uniform_(self.bias, -bound, bound)
19
20              # Define the fused convolution and activation CUDA kernel
21              conv_activation_source = """
22              #include <torch/extension.h>
23              #include <cuda_runtime.h>
24              #include <math.h>
25
26              __global__ void conv_activation_kernel(const float* input, const
        float* weights, const float* bias, float* output,
27                                                      int N, int C_in, int D,
        int H, int W, int C_out, int kD, int kH, int kW,
28                                                      float clamp_min, float
        clamp_max) {
29                  int idx = blockIdx.x * blockDim.x + threadIdx.x;
30                  const int W_out = W - kW + 1;
31                  const int H_out = H - kH + 1;
32                  const int D_out = D - kD + 1;
33                  const int total_elements = N * C_out * D_out * H_out * W_out;
34                  if (idx >= total_elements) return;
35
36                  // Decompose index into output coordinates
37                  int w_out = idx % W_out;
38                  int h_out = (idx / W_out) % H_out;
39                  int d_out = (idx / (W_out * H_out)) % D_out;
40                  int f = (idx / (W_out * H_out * D_out)) % C_out;
41                  int n = idx / (C_out * D_out * H_out * W_out);
42
43                  float sum = 0;
44                  for (int c = 0; c < C_in; ++c) {
45                      for (int kd = 0; kd < kD; ++kd) {
46                          for (int kh = 0; kh < kH; ++kh) {
47                              for (int kw = 0; kw < kW; ++kw) {
48                                  // Input tensor index calculation
49                                  int d_in = d_out + kd;
50                                  int h_in = h_out + kh;
51                                  int w_in = w_out + kw;
52                                  int input_idx = n * C_in * D * H * W +
53                                                  c * D * H * W +
54                                                  d_in * H * W +
55                                                  h_in * W +
56                                                  w_in;
57                                  // Weight tensor index calculation
58                                  int weight_idx = f * C_in * kD * kH * kW +
59                                                   c * kD * kH * kW +
60                                                   kd * kH * kW +
61                                                   kh * kW +
62                                                   kw;
63                                  sum += input[input_idx] *
        weights[weight_idx];
64                              }
65                          }
66                      }
67                  }
68                  sum += bias[f];
69
70                  // Apply activations sequentially
71                  sum = tanh(sum);
72                  sum = fmaxf(clamp_min, fminf(sum, clamp_max));
73                  {
74                      float inner = sum + 0.044715f * sum * sum * sum;
75                      inner *= 0.79788456f; // sqrt(2/pi) approximation
76                      sum = sum * 0.5f * (1.0f + tanh(inner));
77                  }
78
```

```
79              // Output tensor index calculation
80              int output_idx = n * C_out * D_out * H_out * W_out +
81                               f * D_out * H_out * W_out +
82                               d_out * H_out * W_out +
83                               h_out * W_out +
84                               w_out;
85              output[output_idx] = sum;
86          }

87

88      torch::Tensor conv_activation_cuda(torch::Tensor input,
    torch::Tensor weights, torch::Tensor bias,
89                                          float clamp_min, float
    clamp_max) {
90              // Dimension extraction
91              int N = input.size(0);
92              int C_in = input.size(1);
93              int D = input.size(2);
94              int H = input.size(3);
95              int W = input.size(4);
96              int C_out = weights.size(0);
97              int kD = weights.size(2);
98              int kH = weights.size(3);
99              int kW = weights.size(4);

100

101             // Output dimensions
102             int D_out = D - kD + 1;
103             int H_out = H - kH + 1;
104             int W_out = W - kW + 1;

105

106             // Create output tensor
107             auto output = torch::empty({N, C_out, D_out, H_out, W_out},
    input.options());

108

109             // Launch kernel
110             int total_elements = N * C_out * D_out * H_out * W_out;
111             const int block_size = 256;
112             const int grid_size = (total_elements + block_size - 1) /
    block_size;
113             conv_activation_kernel<<<grid_size, block_size>>>(
114                 input.data_ptr<float>(), weights.data_ptr<float>(),
    bias.data_ptr<float>(),
115                 output.data_ptr<float>(), N, C_in, D, H, W, C_out, kD,
    kH, kW,
116                 clamp_min, clamp_max
117             );
118             return output;
119         }
120         """

121

122     conv_activation_header = """
123     torch::Tensor conv_activation_cuda(torch::Tensor input,
    torch::Tensor weights, torch::Tensor bias,
124                                         float clamp_min, float
    clamp_max);
125         """

126

127     # Load the CUDA kernel
128     self.conv_activation = load_inline(
129         name="conv_activation",
130         cpp_sources=conv_activation_header,
131         cuda_sources=conv_activation_source,
132         functions=["conv_activation_cuda"],
133         verbose=True
134     )

135
```

```
136     def forward(self, x):
137         return self.conv_activation.conv_activation_cuda(x, self.weight,
        self.bias, self.clamp_min, self.clamp_max)
```

On turn 2 it attempts to improve on the existing implementation, but fails due to a compilation error.

On turn 3, it attempts to modify the loop order to improve coalesced memory access, but this actually worsens performance, as the kernel now exhibits a speedup of **0.61x**. The model attempts this strategy again at turn 4, and is able to order the loops in a way that increases memory coalescing: this brings up the performance to **1.19x**. At turn 5 and 6, the model attempts some minor optimizations (such as precomputing certain indices, or deleting intermediate variables), which however has no measurable effect in performance.

On turn 7, the model figures out a surprising optimization: it records the convolution's kernel size as a compile time constant, which allows it to use `# pragma unroll` to unroll the 3 loops needed to apply the convolution. This results in a significant speedup of **1.93x** over baseline.

```
1   import torch
2   from torch.utils.cpp_extension import load_inline
3   import math
4
5   class ModelNew(torch.nn.Module):
6       def __init__(self, in_channels, out_channels, kernel_size,
        clamp_min, clamp_max):
7           super(ModelNew, self).__init__()
8           self.clamp_min = clamp_min
9           self.clamp_max = clamp_max
10
11          # Initialize convolution weights and bias
12          self.weight = torch.nn.Parameter(torch.empty(out_channels,
        in_channels, kernel_size, kernel_size, kernel_size))
13          self.bias = torch.nn.Parameter(torch.empty(out_channels))
14          # Weight initialization following PyTorch's Conv3d default
15          torch.nn.init.kaiming_uniform_(self.weight, a=math.sqrt(5))
16          fan_in, _ =
        torch.nn.init._calculate_fan_in_and_fan_out(self.weight)
17          bound = 1 / math.sqrt(fan_in)
18          torch.nn.init.uniform_(self.bias, -bound, bound)
19
20          # Define the fused convolution and activation CUDA kernel with
        optimizations
21          kernel_size_val = kernel_size
22          conv_activation_source = f"""
23          #include <torch/extension.h>
24          #include <cuda_runtime.h>
25          #include <math.h>
26
27          #define KERNEL_SIZE {kernel_size_val}
28          #define KERNEL_D {kernel_size_val}
29          #define KERNEL_H {kernel_size_val}
30          #define KERNEL_W {kernel_size_val}
31
32          __global__ void conv_activation_kernel(
33              const float* __restrict__ input, const float* __restrict__
        weights, const float* __restrict__ bias,
34              float* __restrict__ output,
35              int N, int C_in, int D, int H, int W, int C_out,
36              float clamp_min, float clamp_max) {{
37                  int idx = blockIdx.x * blockDim.x + threadIdx.x;
38                  const int W_out = W - KERNEL_W + 1;
39                  const int H_out = H - KERNEL_H + 1;
40                  const int D_out = D - KERNEL_D + 1;
41                  const int total_elements = N * C_out * D_out * H_out *
        W_out;
42                  if (idx >= total_elements) return;
```

```
43
44                  int w_out = idx % W_out;
45                  int h_out = (idx / W_out) % H_out;
46                  int d_out = (idx / (W_out * H_out)) % D_out;
47                  int f = (idx / (W_out * H_out * D_out)) % C_out;
48                  int n = idx / (C_out * D_out * H_out * W_out);
49
50                  float sum = 0.0;
51
52                  // Precompute constants for GELU
53                  const float sqrt_2_over_pi = 0.79788456f;
54                  const float approx_coeff = 0.044715f;
55
56                  for (int c = 0; c < C_in; ++c) {{
57                      // Compute input channel base pointer
58                      const int input_c_offset = n * C_in * D * H * W + c
     * D * H * W;
59                      const float* input_c = input + input_c_offset;
60
61                      // Compute center position in the input channel
62                      const int center_offset = d_out * H * W + h_out * W
     + w_out;
63                      const float* center = input_c + center_offset;
64
65                      // Weight base for current output channel f and
     input channel c
66                      const int weight_base = f * C_in * KERNEL_D *
     KERNEL_H * KERNEL_W + c * KERNEL_D * KERNEL_H * KERNEL_W;
67
68                      #pragma unroll
69                      for (int kd = 0; kd < KERNEL_D; ++kd) {{
70                          #pragma unroll
71                          for (int kh = 0; kh < KERNEL_H; ++kh) {{
72                              #pragma unroll
73                              for (int kw = 0; kw < KERNEL_W; ++kw) {{
74                                  const int offset = kd * H * W + kh * W +
     kw;
75                                  const float input_val = center[offset];
76                                  const int w_off = weight_base + kd *
     KERNEL_H * KERNEL_W + kh * KERNEL_W + kw;
77                                  const float weight_val = weights[w_off];
78
79                                  sum += input_val * weight_val;
80                              }}
81                          }}
82                      }}
83                  }}
84
85                  sum += bias[f];
86
87                  // Apply activations with optimized order
88                  {{
89                      float tanh_val = tanhf(sum);
90                      float clamped = fmaxf(clamp_min, fminf(tanh_val,
     clamp_max));
91                      float inner = clamped + approx_coeff * clamped *
     clamped * clamped;
92                      inner *= sqrt_2_over_pi;
93                      sum = clamped * 0.5f * (1.0f + tanhf(inner));
94                  }}
95
96                  // Output index calculation
97                  const int output_base = n * C_out * D_out * H_out *
     W_out +
98                                          f * D_out * H_out * W_out +
```

```
99                                         d_out * H_out * W_out +
100                                        h_out * W_out;
101            output[output_base + w_out] = sum;
102        }}

104    torch::Tensor conv_activation_cuda(torch::Tensor input,
    torch::Tensor weights, torch::Tensor bias,
105                                       float clamp_min, float
    clamp_max) {{
106        int N = input.size(0);
107        int C_in = input.size(1);
108        int D = input.size(2);
109        int H = input.size(3);
110        int W = input.size(4);
111        int C_out = weights.size(0);

113        int D_out = D - KERNEL_D + 1;
114        int H_out = H - KERNEL_H + 1;
115        int W_out = W - KERNEL_W + 1;

117        auto output = torch::empty({{N, C_out, D_out, H_out,
    W_out}}, input.options());

119        int total_elements = N * C_out * D_out * H_out * W_out;
120        const int block_size = 256;
121        const int grid_size = (total_elements + block_size - 1) /
    block_size;

123        conv_activation_kernel<<<grid_size, block_size>>>(
124            input.data_ptr<float>(), weights.data_ptr<float>(),
    bias.data_ptr<float>(),
125            output.data_ptr<float>(), N, C_in, D, H, W, C_out,
126            clamp_min, clamp_max
127        );
128        return output;
129    }}
130    """

132    conv_activation_header = """
133    torch::Tensor conv_activation_cuda(torch::Tensor input,
    torch::Tensor weights, torch::Tensor bias,
134                                       float clamp_min, float
    clamp_max);
135    """

137    # Load the CUDA kernel with fast math optimization
138    self.conv_activation = load_inline(
139        name="conv_activation",
140        cpp_sources=conv_activation_header,
141        cuda_sources=conv_activation_source,
142        functions=["conv_activation_cuda"],
143        extra_cuda_cflags=['-use_fast_math'],
144        verbose=True
145    )

147 def forward(self, x):
148     return self.conv_activation.conv_activation_cuda(x, self.weight,
    self.bias, self.clamp_min, self.clamp_max)
```

In its final turn, the model attempts a more advanced implementation that further parallelizes the computation across kernels before performing a warp-level reduction. However, it fails to implement the strategy correctly, due to applying the reduction across the wrong axis. We do note the model has shown success in implementing complex warp reductions in several other tasks.

# I    ERROR CORRECTION BEHAVIOR

An important quality for kernel generation is to correct errors over turns. With multi-turn training, Kevin learns to effectively manage kernel writing across multiple turns, making more aggressive optimizations while correcting errors more effectively. This is shown by our results in Figure 5, where Kevin exhibits better scaling behavior across serial turns (compared to its single-turn counterpart). Here we focus on an example that illustrates error correction behavior.

We consider the KernelBench task of `CosineSimilarityLoss` (Level 1 Task 97):

```
1   # prediction [128, 4096]
2   # target[128, 4096]
3
4   def forward(self, predictions, targets):
5           cosine_sim = torch.nn.functional.cosine_similarity(predictions,
        targets, dim=1)
6           return torch.mean(1 - cosine_sim)
```

Note: the speedups below are calculated with respect to the eager baseline. For this kernel, torch compile causes a slight degradation in performance (0.96x speedup over eager), likely due to additional overhead introduced.

**Test-time Trajectory from Single-Turn RL:**

*Turn 1:* The model implements a correct kernel on the first try. The kernel is straightforward yet performant, obtaining a 3.15x speedup over the reference implementation. This is expected as the single-turn model is trained to maximize single-turn correctness.

*Turn 2:* The model attempts to calculate the block size dynamically; however, it results in suboptimal GPU utilization, causing a decrease in runtime to 2.19x.

*Turns 3-5:* The model reverts the change from Turn 2, and adds minor improvements (such as performing the mean calculation at the warp level rather than using a single thread, or tweaking the loading of shared memory); however, it has no significant impact on performance (the speedups range from 3.11x to 3.19x).

*Turns 6-8:* The model attempts to tune the grid size and block size, though this fails to improve performance (the speedups ranging from 2.99x to 3.14x).

**Test-time Trajectory from Multi-Turn RL:**

*Turn 1:* The model attempts a kernel similar to the single-turn implementation. A suboptimal block size and a non-contiguous access, however, cause a speedup of 1.74x, inferior to the single turn version.

*Turn 2:* The multi-turn model makes various optimizations (such as tweaking block and grid size, and avoiding recomputing certain values), but it forgets to include a required C++ header, causing a compilation error.

*Turn 3:* The multi-turn model adds the correct C++ header and improves the memory coalescing. This, combined with the previous optimizations, causes a speedup of 3.31x.

*Turn 4:* The model now attempts more aggressive optimization techniques to compute the sums within the kernel, namely using warp intrinsics to perform warp-level reductions. The implementation is incorrect, however, as only a subset of the threads for a given warp execute the reduction, resulting in a deadlock.

```
1   for (int s = 8; s >= 1; s >>= 1) {
2       if (thread_id < s) {
3           float temp_dot = __shfl_down_sync(0xFFFFFFFF, dot, s);
4           dot += temp_dot;
5           float temp_p = __shfl_down_sync(0xFFFFFFFF, p_norm_sq, s);
6           p_norm_sq += temp_p;
7           float temp_t = __shfl_down_sync(0xFFFFFFFF, t_norm_sq, s);
8           t_norm_sq += temp_t;
9       }
```

```
10        __syncthreads();
11  }
```

*Turn 5:* The model then tries a different strategy, implementing a simpler reduction that avoids using warp intrinsics. However, this proves less efficient than the previous methods, and achieves a speedup of only 2.26x.

*Turn 6:* The model now attempts to fix the errors present in the advanced warp level technique from turn 3, spending most of the chain-of-thought in trying to address the previous error. This time, the model is successful with a speedup of 7.81x.

```
1  for (int s = 16; s >= 1; s >>= 1) {
2          float temp = __shfl_down_sync(0xFFFFFFFF, dot, s);
3          dot += temp;
4          temp = __shfl_down_sync(0xFFFFFFFF, p_norm_sq, s);
5          p_norm_sq += temp;
6          temp = __shfl_down_sync(0xFFFFFFFF, t_norm_sq, s);
7          t_norm_sq += temp;
8      }
9  ...
```

*Turns 7, 8:* The model tries a few other optimizations, yet has no significant impact on the kernel performance, resulting in slightly worse kernels. The best kernel is thus achieved at turn 6 with a speedup of 7.81x, compared to the best kernel found by single-turn (3.19x).

Comparing the trajectories of the multi-turn vs single-turn model, we notice that the single-turn model is generally careful in its optimizations and rarely deviates from the previous attempts. The multi-turn model attempts riskier strategies. While it may result in wrong kernels sometimes, it is more capable of recovering from these errors and eventually finding more performant ones.

