# OpenReview forum: "Kevin: Multi-Turn RL for Generating CUDA Kernels"
_ICLR.cc/2026/Conference — ICLR 2026 Poster_

### Official Review · Reviewer_dQdu · 2025-10-22

**Soundness:** 2
**Presentation:** 3
**Contribution:** 3
**Rating:** 6
**Confidence:** 3

**Summary:**

This paper introduces "Kevin," a model trained to generate and optimize GPU kernels using a multi-turn RL framework. The study think that the iterative nature of code optimization is best captured by a training process where the model receives and learns from execution feedback over multiple rounds. The core is a multi-turn RL recipe designed to overcome challenges inherent in long-horizon tasks. By explicitly modeling the "generate -> execute -> get feedback -> refine" loop, the training process aligns more closely with how human experts work. Evaluation results on the KernelBench dataset demonstrate that this approach is highly effective, showing that Kevin achieves significant improvements in both the correctness and performance of the CUDA kernels it generates compared to its base model and other strong baselines.

**Strengths:**

1. Well-motivated multi-turn RL framework to solve the Kernel Bench CUDA task. And the approach is validated by strong empirical results. Not only surpasses its base model and a single-turn RL counterpart but also outperforms strong frontier models like OpenAI 04-mini on both correctness and performance metrics.
2. Clearly demonstrate that sequential refinement is more compute-efficient than parallel sampling.
3. Offers practical engineering solution, detailing robust solutions to reward hacking probelm.

**Weaknesses:**

1. The most significant weakness is the reliance on the relatively small KernelBench benchmark, which contains only around 250 tasks. The scarity of training data on this specific domain will raise my concern on the generalizability compared to other large language models.
2. The paper mentioned that "only accurate for those dimentions and NVDIA H200 GPUs". The reported speedup performance improvement are highly specific to H200 evaluation environment within KernelBench. More analysis on the methods potential performance transfer to different hardware architectures or real-world workloads remains an open question, further validation on other benchmark environments are needed.

**Questions:**

Please check weakness.

---

> ### Author Response · Authors · 2025-11-26
> **[1/2] Transferability across hardware and settings**
>
> We appreciate that the reviewer was impressed with “strong empirical results”, such as the effectiveness of sequential refinement over parallel sampling, as well as our novel contribution of the multi-turn RL framework with a robust, “practical engineering solution”, which we think could be a valuable contribution for the community. We address your concerns below and hope these clarifications will assist in your final evaluation.
>
> **Transferability across hardware and settings**
> We appreciate the reviewer’s curiosity about the **potential transfer to different hardware**  platforms or real-world workloads, important to demonstrating Kevin’s usefulness and applicability. We report speedups “only accurate for those dimensions and NVIDIA H200 GPUs”, as we want to rigorously and reliably compute speedups (with H1/200 being a widely used ML training/inference hardware). More importantly, the underlying optimization principles Kevin learned are not inherently tied to these particular settings. Beyond absolute speedup on the benchmark, Kevin demonstrates interesting general behaviors after multi-turn RL: such as error correction (see Appendix I), attempting more aggressive optimizations over turns to achieve higher speedups, and better test-time scaling behavior both serially and in parallel.
>
> Encouraged by your question, we have conducted additional experiments to examine whether these results hold when trained and/or evaluated on other hardware architectures.
>
> First, we present the evaluation results on **NVIDIA A100s**, added to Appendix E.5. Across all metrics, the results demonstrate the same trend as those on H200s, suggesting that our training recipe could generalize to other hardware architectures, despite being trained with execution feedback on H200s. While the focus of this study is to cleanly investigate the design of multi-turn RL in this setting (hence fixing hardware to H200), we could experiment with training the model to be more aware of the hardware architecture in the future, by mixing in kernel evaluation environments from other GPUs.
>
> | On A100 | Correctness |  | Performance |  |
> | :---- | :---- | :---- | :---- | :---- |
> | Method | best@16 | avg@16 | best@16 | avg@16 |
> | Kevin-32B | **79%** | **44%** | **0.88** | **0.33** |
> | Single Turn | 78% | 42% | 0.71 | 0.29 |
> | QwQ-32B | 53% | 12% | 0.45 | 0.08 |
>
> Second, we provide additional evaluations on **KernelBench Level 3 tasks**, as shown in Appendix E.4 (and our response to Reviewer PBZv). These Level 3 tasks require end-to-end optimization of full and realistic model architectures, such as the VisionTransformer, and miniGPT attention blocks; these vary significantly from the shorter PyTorch programs Level 1 and 2 Tasks (on which Kevin was trained), composed of one or a few basic operators. These Level 3 results show that multi-turn RL can generate faster kernels for these much more complex tasks over both single-turn RL and the base model, highlighting our approaches’ effectiveness and generalizability on new tasks and shapes.
>
> Finally, we are glad the reviewer brought up a point about **potential transfers to real-world workloads** (beyond benchmark results). We propose several potential ways that Kevin could assist kernel developers:
>
> 1. For a given task, developers may spin up Kevin in the background to generate a large number of kernels and explore an array of optimization strategies, rather than being forced to only focus on a small subset of possible optimizations due to time constraints. Generating a single sample is relatively inexpensive, making Kevin suitable for extensive use of test-time computation.
> 2. Kevin’s ability to customize for each problem’s specifics allows for more aggressive optimizations targeted to the user’s configuration (such as tensor sizes, underlying hardware, …).
> 3. Since Kevin is trained to correct errors and refine kernels, developers may also let Kevin improve existing incorrect or suboptimal kernels.
>
> We’d like to note that while we open-source our model and training recipe, creating an interface for developers or deploying such systems is beyond the scope of this study; yet we hope our multi-turn RL approach could eventually enable such agentic assistants in the real world.

---

> ### Author Response · Authors · 2025-11-26
> **[2/2] Generalizability & Training Tasks**
>
> **Generalizability & Training Tasks**
> We agree with the reviewer that there is a limited number of tasks for this benchmark. As discussed in our response to reviewer 5ts9, constructing **robust, high-quality environments** for kernel generation is a significant challenge for the community and beyond the scope of this paper.
>
> Regarding the reviewer’s concern about the **generalizability** of our method, we have conducted additional experiments across new dimensions. In the above response, we included additional evaluation results on NVIDIA A100s (Appendix E.5), a different GPU architecture than the one used during training (H200). Our results on the more challenging and unseen KernelBench Level 3 problems (Appendix E.3) suggest that Kevin could solve complex, out-of-distribution problems. In response to reviewer PBZv, we’ve shown that our results are consistent when run with the torch.compile baseline (Appendix E.4).
>
> **Qualitatively**, we also observe during training that once the model learns to effectively use a given technique (such as warp shuffling) on a given problem type (such as matrix multiplications), it’s able to then quickly apply it to another class of problems (such as convolutions) in subsequent turns. We’ve additionally provided an example in Appendix I, where Kevin demonstrates improved capability in correcting errors over turns, an optimization strategy useful for all problems across KernelBench and kernel optimization in general.
>
> While we acknowledge that the model would likely not generalize well to domain-specific languages other than CUDA (such as Triton or CUTLASS), we emphasize that Kevin could enable **generalization within the CUDA kernel** domain. As shown in papers such as \[1\], RL training can improve performance significantly even with a small dataset, as each sample effectively allows the model to receive feedback from exploring different approaches. Concretely, Kevin’s multi-turn RL explicitly boosts this by encouraging exploration through multiple turns during rollout. Thus, each environment can serve as a source of multiple diverse rollouts with different optimization trajectories and varying end performance, enabling us to extract more training signals per task and providing sample-efficiency gains. We can envision a future version of the work that addresses the scarcity limitation by maximizing performance in a single environment, in a similar spirit to test-time training \[2\] approaches.
>
> \[1\] Reinforcement Learning for Reasoning in Large Language Models with One Training Example
> \[2\] Learning to (Learn at Test Time): RNNs with Expressive Hidden States

---

### Official Review · Reviewer_cuJN · 2025-10-28

**Soundness:** 2
**Presentation:** 2
**Contribution:** 2
**Rating:** 4
**Confidence:** 3

**Summary:**

The paper proposes Kevin, a CUDA kernel generation model trained with multi-turn RL. Instead of producing one kernel in one shot, the model iteratively refines kernels over several turns, using runtime and profiling feedback from previous attempts. The training pipeline assigns discounted reward across turns and treats each refinement step as its own RL sample, with context summarization to keep prompts short.
On 100 held-out CUDA tasks (KernelBench style), Kevin matches a strong single-turn RL baseline in correctness (82% best@16) but shows somewhat higher peak runtime speedup (1.10× vs 0.85× best@16).
The paper argues that this proves multi-turn RL improves the ability to iteratively optimize kernel performance and scales better when you allow more refinement turns at test time.

**Strengths:**

**Realistic problem setup.** The work models how GPU performance engineers actually iterate: propose kernel → profile → refine. The RL formulation explicitly credits early partial attempts that later lead to a fast kernel, instead of rewarding only final outputs.

**Clear engineering advances.** They introduce (i) per-turn training on every refinement step to improve sample efficiency and (ii) discounted future-return style reward aggregation for credit assignment across turns.

**Beating strong general models.** Kevin, post-trained from QwQ-32B, outperforms powerful proprietary baselines like o4-mini and o3-mini on this CUDA kernel benchmark, even though those models are otherwise stronger than QwQ-32B on standard coding and reasoning tasks.

**Test-time scaling insight.** The paper studies how performance scales with more refinement turns and shows the multi-turn–trained model benefits more from extra turns than single-turn RL, suggesting better iterative optimization ability.

**Weaknesses:**

**Performance gains are marginal vs the main ablation.** Compared to the single-turn RL baseline, Kevin does not improve solve rate: correctness best@16 stays 82% vs 82%, and fast1 best@16 stays 43% vs 43%. The main improvement is higher best-case speedup (1.10× vs 0.85×), and average speedup only rises slightly (0.40× vs 0.35×).
So the method is not solving more tasks; it’s mostly squeezing somewhat better runtime on tasks that were already solvable.

**Heavy system heuristics, limited theory.** The approach bundles many practical tricks (turn-by-turn training, summarizing CoT to avoid 50k–100k token contexts, discounted reward shaping).
But these are presented as an engineering recipe rather than a principled analysis of why multi-turn RL should outperform single-turn RL.
Detailed ablation study would help us understanding real contribution of introducing multi-turn.

**Questions:**

Q1: Multi-turn RL and single-turn RL have the same solve rate (82% best@16) and same fast1 rate (43%), and only a modest speedup gain (1.10× vs 0.85×). In what concrete situations does multi-turn RL succeed where single-turn RL fails (either solving a task it couldn’t solve, or finding a faster kernel it couldn’t reach)? Please quantify.

Q2: Kevin is trained starting from a strong 32B base model. You mention weaker bases tend to hack rewards or collapse.
Can you show any numbers (even partial) for smaller models, or describe concretely how and why training fails there?

Q3: Your method bundles many heuristics (CoT summarization, discounted reward shaping etc). Can you provide ablations that isolate the effect of multi-turn RL itself versus these stabilizing heuristics? Without that, it’s hard to attribute the reported gains to “multi-turn” rather than to engineering tricks.

---

> ### Author Response · Authors · 2025-11-26
> **[1/2] Performance Interpretation and Multi-Turn RL Behavior**
>
> We appreciate reviewer CuJN for recognizing our training recipe as insightful and clear, introducing advances on a “realistic problem setup”, outperforming “strong general model” while revealing “test-time scaling insights” with our multi-turn RL training. We would like to take the opportunity to provide more details and context; we hope these details address your concerns, and would be grateful if you would consider this in your evaluation.
>
> **Interpreting Performance Gains**
> We thank the reviewer for closely looking at our speedup results in Table 1, and we like to provide more context for the results. For GPU optimization, we always strive for finding the **fastest correct kernel**; hence, best-case speedup (best@16) is a much more **indicative** and useful metric than average-case speedup (avg@16), which we report for completeness.
>
> Given that our primary focus is generating fast kernels, we’d argue that the best@16 performance gain of multi-turn over single-turn RL (1.10x vs 0.85x) across all problems is **significant**. We would like to highlight the confidence intervals reported in Appendix E.1, showing the 1.1x performance improvement to be statistically significant and robust. To provide more context about realistic speedups in our setting, squeezing performance on a GPU kernel is hard, especially against the highly optimized PyTorch baselines (which launch kernels hand-written by expert engineers). Hence, even achieving small speedups is significant; large speedups are rare and generally an indication of reward hacking (Appendix G) as indicated by learnings in the community \[1\]\[2\]. To avoid reward hacking, our generation setting explicitly disallows any PyTorch operators from appearing in the final generated kernels; only pure CUDA code is allowed.
>
> The reviewer makes an excellent observation about the similar **correctness rates**. We'd like to offer some intuition for why single-turn and multi-turn both achieve 82% best@16 correctness. With single-turn RL, as model rollouts only have one turn, it is incentivized to immediately output correct kernels and avoid risky optimizations, which explains single-turn’s high correctness rate. In contrast, with multi-turn RL training, the model is able to learn how to perform more aggressive optimization over turns; while such aggressive optimizations risk breaking correctness, our multi-turn training demonstrates substantial **speedups without sacrificing accuracy**, striking a balance between this correctness-performance tradeoff.
>
> \[1\] Towards Robust Agentic CUDA Kernel Benchmarking, Verification, and Optimization
> \[2\] BackendBench: An Evaluation Suite for Testing How Well LLMs and Humans Can Write PyTorch Backends
>
> **Concrete Situation of Multi-Turn RL vs Single-Turn RL**
> We appreciate the reviewer’s curiosity in understanding behavioral differences in multi vs single-turn generated kernels. We would like to point the reviewer to our **qualitative analysis** on multi-turn kernel optimization behavior in Appendix H and a comparison between single- and multi-turn trajectories in Appendix I. Concretely, Kevin is more likely to attempt aggressive optimizations and more capable of recovering from the resulting errors.
>
> Quantitatively, our results on best@16 and avg@16 performance with confidence intervals (Appendix E.1) demonstrate that Kevin produces faster kernels on average across the dataset. Moreover, Kevin achieves a higher fast\_1.5 metric, suggesting that it's better at finding kernels with significant speedups.
>
> **Training Behavior on Weaker Priors**
> We have extended Appendix B.6 to elaborate on our choice of the base model in detail.  The key challenge is that other similar open-source base models have significantly weaker priors than QwQ-32B on reasoning tasks like math and coding. This shows up in the training behavior: when trained with Gemma-27B, the **sparse reward signal** often gives zero gradients, preventing the model from improving on this task. We observe these training difficulties on weaker base models, where the model either fails to learn meaningful optimizations or resorts to reward hacking (described in Appendix G). However, we believe that as more and stronger open source models emerge from the community, more models will become viable for this multi-turn RL setting.

---

> ### Author Response · Authors · 2025-11-26
> **[2/2] Contribution of Kevin's Training Recipe**
>
> **Contribution of Kevin’s Training Recipe**
> We appreciate the reviewer's recognition of our “clear engineering advances" for multi-turn RL. We would like to emphasize that we arrived at our recipe after **a** **series of principled analyses** towards designing and applying multi-turn RL on practical engineering tasks, rather than ad-hoc heuristics; each design choice was motivated by necessity and **validated through systematic ablations.**
>
> **Multi-turn Outperforms Single-Turn:**
> Throughout our study, we motivate and demonstrate why multi-turn outperforms single-turn in our problem setting. Kernel engineering (and more in general, code optimization) is inherently an iterative task, which **benefits from multiple interactions** with the environment. As shown in Figures 2, 4 and our analysis in Sections 3.3, 4.4, during RL training, multi-turn shows steady improvement across training steps (as opposed to single-turn, where the reward quickly plateaus). Recent state-of-the-art open source models \[1\]\[2\]\[3\] are similarly post-trained with multi-turn RL in agentic environments, highlighting the value of multi-turn RL for code generation and agentic tasks. Enabling multi-turn RL in this setting required addressing core challenges. Our study systematically studies (1) context management under length constraints and (2) credit assignment across turns. Below, we detail how these challenges were addressed through principled design.
>
> **Context Management**
> To enable incorporating multiple turns during rollout, we must address this practical constraint as QwQ-32B's 32k token context window (without RoPE extension) accommodates only 1-2 full turns, and each turn with CoT requires \~16k tokens. Our CoT summarization (Section 4.1) approach is necessary and **effectively compresses the context** and critical CoT details while fitting in context length. As Reviewer 5ts9 points out, context management will grow more challenging with longer interactions; our method provides a simple yet applicable solution to this challenge in the multi-turn RL setting.
>
> **Credit Assignment across Turns**
> Attributing rewards across multi-turn interactions is a key question we systematically investigate. Our approach expands upon existing strategies \[4\] that use outcome-only rewards for multi-turn trajectories by leveraging per-turn discounted rewards to **provide denser learning signals**. We arrived at our final reward design after thorough analysis in our multi-turn reward design (Section 4.2, 4.3): we treat each turn as a training sample and explore multiple approaches for multi-turn credit assignment, in particular greedy outcome (zero discount factor), maximum (selecting best outcome), and discounted sum (standard RL formulation). We provide both intuition and empirical validation: greedy rewards fail to credit early suboptimal turns that ultimately lead to high-performance kernels (Appendix B.1), and we note discounted sum performs best for our setting (as it credits incremental improvements across turns). We provide these detailed ablations across these designs in Section 4 and Appendix B.
>
> All of Kevin’s design choices that the reviewer listed, *turn-by-turn training, summarizing CoT to avoid 50k–100k token contexts, discounted reward shaping*, are designed in a **general solution to key challenges to multi-turn RL** rather than heavily heuristics specifically based on the kernel generation setting. Section 2.2 covers theoretical work describing multi-turn RL’s advantages \[5\] and empirical turn-by-turn training in other domains \[6\]. Beyond these, we propose novel proxy signals to address training instability and effective reward hacking mitigations for code optimization. We believe our **careful study of RL Training dynamics** on real-world agentic coding tasks, like those elaborated in concurrent work \[7\]\[8\], provides actionable and valuable insights for the broader RLVR community.
>
> \[1\] Kimi K2: Open Agentic Intelligence.
> \[2\] GLM-4.5: Agentic, Reasoning, and Coding (ARC) Foundation Models
> \[3\] Qwen3 Technical Report
> \[4\] RLEF: Grounding Code LLMs in Execution Feedback with Reinforcement Learning
> \[5\] Optimizing Test-Time Compute via Meta Reinforcement Fine-Tuning
> \[6\] Synthetic Data Generation & Multi-Step RL for Reasoning & Tool Use
> \[7\] CWM: An Open-Weights LLM for Research on Code Generation with World Models
> \[8\] INTELLECT-2: A Reasoning Model Trained Through Globally Decentralized Reinforcement Learning

---

### Official Review · Reviewer_PBZv · 2025-10-31

**Soundness:** 1
**Presentation:** 1
**Contribution:** 2
**Rating:** 2
**Confidence:** 4

**Summary:**

Kevin is a multi-turn reinforcement learning (RL) based framework to train LLMs to generate better GPU kernels. Authors propose a design for multi-turn RL with a reward function proportional to correctness and generated kernel's performance. Authors have used GRPO for training the model. During training authors sample m parallel trajectories with n iterative refinement steps. Authors also propose to compress CoTs from prior attempts to save on context length available for generation. Authors have demonstrated that single-turn RL saturates early on and does not benefit from the refinement aspects for test-time compute scaling. Authors also propose to training on each turn in the multi-turn realization. Authors aggregate the reward across multiple turns with two specific discount factors 0.4 and 0.8. Multi-turn training behaviour shows reward monotonically increasing on an average. Authors have used 180 tasks out of 200 (from level 1 and 2) tasks from kernelbench benchmark. Remaining 20 and additional author created 80 tasks are used for evaluation. Authors compare speedup across pytorch implementation provided in tasks. The paper also shows that test-time scaling trends outperform the baseline LLM scaling. Further, authors have shown that Kevin benefits sequential axis of test-time scaling than that of parallel.

**Strengths:**

- Demonstration of multi-turn RL for GPU kernel generation.
- RL reward formulation as a function of correctness and speedup.
- Effectiveness of multi-turn RL to improve inference time scaling trends.
- Clearly comparing gains against single-turn RL.

**Weaknesses:**

- The evaluation methodology does not follow standard practices. Kevin trains on 180/200 examples from kernelbench evaluation benchmark. The authors must create their own dataset for training and then evaluate on kernelbench. This does not inspire fair comparison with existing approaches and goes against standard practices.
- Paper does not specify where are the initial CoTs obtained from.
- In section 4 line 237, the description is unclear.
- In subsection 4.1: Summarizing all previous CoTs does not appear to be the best way to build context. There are several prior methods that have shown using prior top-k + bottom-k attempts to build context that results in better contrastive learning.
- subsection 4.2 appears to be incomplete.
- Performance is evaluated with Pytorch implementation as a baseline. However, even if the benchmark provides reference Pytorch implementations, these are not truly optimal implementations available today. I encourage authors to compare speedup against torch.compile as their baseline. This will show true improvements over manual kernel optimization which is a very time consuming and tedious task. This will justify all the effort and energy spent in training/inference/generation of LLMs.
- There is no profiler feedback integrated. Neither during training nor during inference. The profiler feedback is very crucial which helps human experts to reason and write best strategies to improve kernel performance.
-  Authors compare against o3/4-mini models which are not optimized for code. They should instead compare against GPT4.1 and claude-sonnet-4 which are optimized for code.
- With best@16 Kevin shows 1.10x performance improvement. Which does not inspire any confidence in the efficacy of Kevin given that 10% gain could come from measurement errors.
- There are some case studies in appendix showing speedup of 1.9x, 2+x, 3+x; all these speedups are over naive pytorch calls. torch.compile significantly optimizes a given pytorch code by performing operator fusion on the fly using a pool of manually optimized kernels. This should be the ideal baseline that AI solution must beat.

**Questions:**

Please see weaknesses section.

---

> ### Author Response · Authors · 2025-11-26
> **[1/4] Context + Feedback Construction**
>
> We thank Reviewer PBZv for their detailed and thoughtful comments. We are encouraged that you found our multi-turn RL demonstration interesting and effective, and we greatly appreciate the thoroughness of your suggestions, which motivated several valuable new experiments. In particular, we evaluated Kevin’s generalization on KernelBench Level 3 and added comparisons with torch.compile and other frontier models. We hope these clarifications and additional insights provide helpful context for your final evaluation.
>
> **Context \+ Feedback Construction**
> We thank the reviewer for paying attention to our CoT summary, and we agree that effectively constructing contexts is indeed critical in the multi-turn setup. As noted in Section 4.1, naively keeping the previous full chains-of-thought (CoTs) quickly exhausts the context window. Our method takes notice that the CoT summaries instead are only a few hundred tokens each, making it **feasible to retain summaries** from all previous attempts within our 4-turn setup.
>
> The reviewer’s suggestion of building context with top-k and bottom-k attempts is nonetheless interesting, and could be a promising approach as we scale to more serial turns and models having longer context length. For our current setup, we make the model output the CoT summary and **keep all of them in context**. This allows the model not only to write performant kernels, but also to learn which information is useful to store in the CoT summary for future turns (as opposed to predefined heuristics that constrain the model more).
>
> We appreciate the reviewer's suggestion to incorporate profiling feedback; we believe incorporating feedback during training / inference could help improve kernel generation. We explored our multi-turn RL reward to rely **solely on end metrics** (correctness and wall clock runtime), reducing the need for heuristics to select intermediate signals to incorporate. This makes our methodology more generalizable, including settings where profilers may not be available or practical. We see this as a strength of our approach and potential to apply our method to other multi-turn optimization tasks beyond kernel generation.
>
> That being said, we recognize that leveraging the profiler presents some **interesting opportunities and technical challenges** to be used effectively:
>
> - Context constraints: Profiling a kernel on H200 with an NVIDIA Nsight Compute (NCU) would generate \~2500 unique hardware counters, quickly filling up the context window and potentially confusing the model.
> - Reward Heuristic Design: Incorporating profiler metrics into kernel-score design would require heuristics to choose which information to display in the context.
> - Reward Hacking risk: Optimizing for specific proxy metrics (e.g., cache utilization, occupancy) might lead the model to hack these signals without improving wall-clock performance, which is our ultimate objective.
> - Hardware Variability: GPU architecture varies across generations and has some different profiling counters, restricting generalizability across platforms.
>
> Nonetheless, we thank the reviewer for the suggestion. We think our outcome-based approach could be a foundation to carefully explore integrating with profiler signals, with the additional complexity. The goal of Kevin is to demonstrate the effectiveness of multi-turn RL on **solely the outcome reward** metric (correctness and speedup).

---

> ### Author Response · Authors · 2025-11-26
> **[2/4] Evaluation Methodology**
>
> **Evaluation Construction**
> We appreciate the reviewer's careful attention to our evaluation methodology. We address this concern by clarifying our approach and providing additional evaluation results.
>
> **KernelBench as an Optimization Environment**
> We agree with the reviewer that fair comparison with "existing approaches” for evaluation is important, and we appreciate the opportunity to clarify our evaluation methodology. KernelBench was explicitly designed as “a real-world engineering environment” according to the original paper \[1\], and thus intended as an **optimization environment** rather than a static benchmark with train/test split. Various efforts from the community \[2\]\[3\]\[4\] (cited in Section 2.1) use KernelBench Level 1+2 tasks as environments for agentic optimization or evolutionary search-based approaches, **where these optimization strategies have iterative access to the KernelBench environment itself.** Our RL-based approach mirrors those strategies and treats a subset of these tasks as RL environments (after improvements as explained in Appendix A.1). This is consistent with other multi-turn RL work \[5\] that augments existing benchmarks and splits into train / eval environments. We acknowledge that we are limited by the scarce availability of high-quality environments for CUDA performance optimization (as shown in KernelBench and TritonBench), and curating additional environments is a very delicate and time-intensive process (to avoid reward hacking). We are excited about future research that can scale up high-quality kernel environments, which would enable RL approaches like Kevin to scale.
>
> We also agree with the reviewer that evaluating our RL approach outside of the training environment is important: this is why we ensure **no train/test contamination** through our evaluation set, that they are completely held out and unseen during training. All of our evaluations and comparisons across models are done on **this identical held-out set** (following the methodology described in Appendix A.2). With this setup, our paper then focuses primarily on developing a robust and effective recipe for multi-turn RL training.
>
> \[1\] KernelBench: Can LLMs Write Efficient GPU Kernels? Section 1
> \[2\] Towards Robust Agentic CUDA Kernel Benchmarking, Verification, and Optimization
> \[3\] Measuring Automated Kernel Engineering, [MET](https://metr.org/)R
> \[4\] GPU Kernel Scientist: An LLM-Driven Framework for Iterative Kernel Optimization
> \[5\] Scaling Long-Horizon LLM Agent via Context-Folding
>
> **Demonstrating Generalization on Unseen KernelBench Tasks**
> To further validate that our approach learns generalizable optimization strategies rather than overfitting to the training environment, we also conducted evaluation on the **completely unseen KernelBench Level 3** (See Appendix E.3). These Level 3 tasks require end-to-end optimization of full model architectures, such as the VisionTransformer, and miniGPT attention blocks; these vary significantly from the Level 1 and 2 Tasks (on which Kevin was trained), composed of one or a few basic operators.
>
> As shown in the table below, multi-turn RL can generate faster kernels for these much more complex tasks over both single-turn RL and the base model, highlighting our approaches’ effectiveness and **generalizability**.
>
> |   | Correctness |  | Performance |  |
> | :---- | :---- | :---- | :---- | :---- |
> | **KernelBench Level 3** | best@16 | avg@16 | best@16 | avg@16 |
> | **Multi-turn RL** | **36%** | **11.75%** | **0.41x** | **0.08x** |
> | Single-turn RL | 36% | 8.38% | 0.36x | 0.06x |
> | QwQ-32B | 4% | 0.25% | 0.04x | 0.002x |

---

> > ### Author Response · Authors · 2025-11-26
> > **[3/4] Baseline Model Choice**
> >
> > **Baseline Model Choice**
> > We thank the reviewer for suggesting more frontier models for comparisons, namely GPT4.1 and claude-sonnet-4, that the reviewer mentioned are "optimized for code”. We first note that our existing baseline models (o3/o4-mini) **demonstrate very strong coding capabilities** (per [o3-mini release post](https://openai.com/index/openai-o3-mini/)) mentions, the model has “exceptional STEM capabilities—with particular strength in science, math, and coding—”. In leading benchmarks such as SWE-bench verified, o3-mini and o4-mini achieve [61%](https://openai.com/index/openai-o3-mini/) and [68%](https://openai.com/index/introducing-o3-and-o4-mini/) respectively, compared to GPT-4.1, which achieves 55%.
> >
> > As per reviewer's request, we run our evaluation harness on both GPT4.1 and Sonnet-4. The results are as follows:
> >
> > | Model      | Correctness best@16 | Correctness avg@16 | Performance best@16 | Performance avg@16 | Fast@1 best@16 | Fast@1 avg@16 | Fast@1.5 best@16 | Fast@1.5 avg@16 |
> > |-----------|----------------------|---------------------|----------------------|---------------------|-----------------|----------------|-------------------|------------------|
> > | Kevin-32B | **82%**              | **46%**             | **1.10x**            | **0.40x**           | **43%**         | **15%**        | **20%**           | **6%**           |
> > | Sonnet-4  | 71%                  | 34%                 | 0.69x                | 0.26x               | 25%             | 8%             | 7%                | 3%               |
> > | GPT 4.1   | 36%                  | 14%                 | 0.25x                | 0.09x               | 8%              | 3%             | 4%                | 2%               |
> >
> > We observe that Kevin still shows **improvements** over these frontier models. Interestingly, we note that compact reasoning models like o4-mini perform comparably / better on kernel generation compared to Sonnet-4 and GPT-4.1, highlighting that this is a task that strongly benefits from extended reasoning and justifying our original selection of frontier reasoning models for comparisons.

---

> > > ### Author Response · Authors · 2025-11-26
> > > **[4/4] Performance Significance and torch.compile Comparison**
> > >
> > > **Performance Significance**
> > > We would like to highlight 1.1x best@16 performance improvement on CUDA is **significant in the context of CUDA kernel authoring.** To answer the concern about 10% coming from measurement error, we would like to point the reviewer to the confidence intervals reported in Appendix E.1, showing the 1.1x performance improvement to be statistically significant. In addition, each kernel's speed is measured 100 times following KernelBench to ensure accurate estimation, yielding very small standard deviations (with std typically \<0.1% of mean across trials) that minimize measurement noise.
> > >
> > > Furthermore, we note that the best@16 1.1x is computed as **average across tasks** (including the tasks where the Kevin-32B written CUDA kernel underperforms the vanilla torch implementation). As the fast@p metrics show, we can see how Kevin achieves a speedup greater than 1.5x in 20% of the kernels. Furthermore, the default PyTorch implementations for many of the KernelBench tasks (especially Level 1\) are already extremely optimized with hardware and shape-specific kernels written by experts, making the speedup significant.
> > >
> > > **Performance Against Torch Compile**
> > > In the original submission, we followed KernelBench and many follow-up works \[1\]\[2\]\[3\] and computed speedup over PyTorch eager. As reported in \[1\], the support and speedups gained by torch compile vary heavily across hardware platforms, making it less suitable for standardized benchmarking compared to its eager counterpart.
> > >
> > > However, we recognize torch.compile might be a **stronger baseline** to compare against, especially on our evaluation set (see construction in Appendix A.2). We’ve added the results in Appendix E.4, which demonstrate that multi-turn RL **still achieves higher best@16** performance (from 1.10x against eager to 1.04x against compile) and a similar avg@16 performance compared to single-turn RL.
> > >
> > > \[1\] MultiKernelBench: A Multi-Platform Benchmark for Kernel Generation
> > > \[2\] Towards Robust Agentic CUDA Kernel Benchmarking, Verification, and Optimization
> > > \[3\] Speeding up pytorch inference on apple devices with AI generated kernels
> > >
> > > We have also updated the case study in Appendix H and Appendix I to include the torch compile number for the trajectory. We note the following:
> > >
> > > - For the sample kernel in Appendix H, compiling the kernel achieves a 1.32x speedup over the eager version. However, our model outperforms this baseline, with a speedup of 1.46x over the compiled kernel.
> > > - For Appendix I, compilation actually causes slight performance degradation, with a 0.96x speedup over the eager version (this is due to the additional launch overhead introduced by torch compile).
> > >
> > > **Other Feedback**
> > > Thank you for carefully examining our study and pointing out potential sources of confusion. We have updated our paper and offer **more clarification** below.
> > >
> > > * *Re: obtaining Initial CoT \-*  Given that QwQ-32B (the base model for Kevin) is a reasoning model (having gone through reasoning post-training already), it is able to generate chains of thought on its own.
> > > * We have expanded and updated the description in Section 4 regarding how we construct samples and rewritten Section 4.2 for additional clarity.

---

### Official Review · Reviewer_5ts9 · 2025-11-03

**Soundness:** 3
**Presentation:** 3
**Contribution:** 3
**Rating:** 8
**Confidence:** 3

**Summary:**

In this paper, the authors propose an RL training method to generate high-performance CUDA kernels by using multi-turn RL training, known as Kevin. In contrast with previous code-generation studies that focused only on correctness, Kevin starts from a reference PyTorch implementation and lowers it to CUDA, monitoring both correctness and performance. To accomplish this, the authors assign a score to each kernel that balances correctness and performance, and train an LLM using GRPO on samples without immediately iterating on external feedback, as is typical of single-turn training. Instead, during each training step, multiple responses are evaluated and assigned a reward. During multi-turn training, not only is the code used to stir the process, but the chain of thought as well. However, with the aggregation of so many instances, the CoT context becomes quite large, so the authors summarize the context from earlier runs. Two approaches to assigning scores were investigated: a greedy approach that assigns a kernel score to each turn and an outcome-based approach that assigns a score to all turns based on the best score in the trajectory. The authors propose a hybrid approach that balances the pros of these two credit assignment strategies and perform an ablation study to support the efficacy of this choice. The evaluation section illustrates the improved performance of multi-turn training compared to single-turn alternatives and other LLMs to generate code. The code generated by KL yields a higher rate of correctness and achieves improved performance in most tests.

**Strengths:**

- This is my first time encountering multi-turn in the context of code generation. The results support its usage from both a correctness and performance perspective.
- The authors investigated multiple aspects of the RL training procedure, from the generation of the training data, reward assignment, and the composition and structure of samples during training. All of these training hyperparameters were shown to have a notable impact on the quality of the result and should provide a useful data point for other researchers in the area, considering multi-turn RL.
- The issue of context length control seemed to be an interesting, but maybe ephemeral problem, as the context lengths continue to grow. The solution they proposed did not seem detrimental to kernel improvement.
- The tradeoff inference configurations, illustrated in Table 2, were an interesting display of the performance impact of trajectory vs turns on both the performance and correctness.

**Weaknesses:**

- I found the discussion of the choice of baseline model in Appendix B.6 to be insufficient from the reader's perspective. While it seems completely plausible for the largest model to have the best priors and smaller models more susceptible to reward hacking, it may be the case that certain updates to the reward function or training could alleviate these issues.
- One of the major limitations, noted by the authors, is the limited number of robust tasks usable for training. With access to more tasks, the kernel generation capabilities of Kevin could be significantly higher.
- All the figures are difficult to digest for people who struggle with differentiating colors. A change in the markers and/or line styles would make it easier to tell the differences at a glance.

**Questions:**

- Is it possible to add a graph to illustrate the issues related to smaller models and reward hacking in Appendix B.6? Not strictly necessary, but it would provide more context for readers who are not as familiar with these issues.

---

> ### Author Response · Authors · 2025-11-26
>
> Dear Reviewer,
>
> Thank you for your thoughtful review. We are glad that you appreciate the novelty of multi-turn training applied to performance optimization tasks, and our thorough analysis of the RL training procedure and demonstrating interesting tradeoffs. We agree with your opinion that “context length control” may be an "ephemeral problem", as context length increases; we designed our context management to be general so that it could be applied beyond kernel generation.
>
> **Choice of Base Model**
> We appreciate the reviewer's concern about the choice of base model for RL training. Through our attempts (Appendix B.6), we found that a certain degree of **capability is needed for the base mode**l (such as QwQ-32B for us), otherwise, they (smaller models) cannot generate the learning signal necessary for RL to succeed. Across the base model we attempted, QwQ-32B is  significantly stronger in terms of math and coding capability (see Table 3, Appendix B.6 in the updated paper).
>
> Given that gradient updates are only possible if the model receives a **non-zero / non-sparse reward** in any of the rollouts for a given task, and that for many kernels a weak model is unable to generate any correct solutions, the model only receives a very sparse signal for the few kernels that it’s able to implement correctly. This, however, is not sufficient for the model to learn, hence why, even after 15 steps, the reward shows no improvements. This is reflected in the attempted training runs with these models: with the same recipe, there is no improvement in performance even after dozens of steps, with the reward staying flat.
>
> The reviewer makes an interesting point that reward function design could potentially address capability gaps. We actually experimented along this direction, hoping **intermediate rewards** might help smaller models learn. In particular, we explored in Section 3.2 rewarding  intermediate steps (such as correct compilation) to increase signal density, but that caused the model to overoptimize for these intermediate steps and not actually learn to implement correct and performant kernels.
>
> As reviewers are curious about other base model’s **training dynamics**, we have updated Appendix B.6 to show the training reward curve for Gemma-27B per your request. We can see that the reward signal is extremely sparse, and the model is not able to improve, given the zero gradient for the majority of training steps. Encouragingly, we believe that as stronger open-source models become available at smaller scales, a wider array of smaller models with strong priors could be leveraged for successful RL training in this domain.
>
> **Limited Numbers of Tasks**
> We acknowledge there is a scarcity of **available high-quality training environments** for the kernel generation domain; KernelBench is the most widely used such collection of environments. We attribute the scarcity of environments to the complexities of constructing them; as discussed in Appendix A.1, they have to be carefully examined to prevent reward hacking, and we spent a significant amount of time improving the KernelBench tasks. Constructing more kernel environments at scale is outside of the scope of this study, but we believe that as the community creates more such environments, Kevin’s multi-turn RL strategy would scale even better.
>
> **Figure Style**
> We will update the figure style with line markings for our camera-ready version, per your recommendation.

---

### Author Response · Authors · 2025-11-26
**Summary of Reviews and Our Rebuttals**

Dear Reviewers and Area Chairs,

Thank you for your time and dedication to reviewing our paper and for your detailed feedback. We are glad that our reviewers found Kevin insightful and interesting for:

1. **Well-motivated and novel application** targeting “realistic problem setup” of KernelBench-style CUDA kernel generation and a pioneering work of leveraging “multi-turn \[RL\] in the context of code generation”.
2. **Interesting multi-turn RL design** with reward as a “function of correctness and speedup” balancing tradeoff of this multi-objective task, and “per-turn” rewards improve sample efficiency; and our “generate \-\> execute \-\> get feedback \-\> refine" training loop is aligned with “how human experts work.”
3. **Strong empirical results** with multi-turn RL showing “clearly .. gains” over single-turn and “outperforms powerful proprietary baselines", and how RL “improve inference time scaling trends”.
4. **Comprehensive technical investigations** for multi-turn RL Training across “multiple aspects”, providing “practical” and “robust solutions” to various challenges from denser reward, training stability, managing context lengths and reward hacking, providing “useful data points for other researchers.. in multi-turn RL.”

Your feedback and suggestions directly inspired several new experiments that we're excited to share. We hope they address your specific concerns while also enabling better understanding of Kevin's behavior and our method. Below, we highlight three key themes shared across reviewers:

- **Additional Evaluation focusing on Generalization:** We thank the reviewers for their valuable suggestions and conducted additional experiments showing that Kevin **generalizes** to completely unseen and challenging KernelBench Level 3 tasks and to other GPU platforms beyond H200. When tested against additional models (Claude Sonnet 4, GPT-4.1) and torch.compile baselines as suggested, **Kevin maintains its advantage over single-turn RL and frontier models**. We also provide important context on how we leverage KernelBench tasks as RL environments, and that achieving speedup on CUDA kernels is very difficult, with 1.1x speedup representing significant gains.
- **Design of multi-turn RL:** We provide more specifics on our design rationales with context construction through CoT summary and focusing on end-to-end metrics (performance and speedup) rather than specific profiler metrics, aiming to create robust and generalizable methods. We arrived at these designs through systematic study and ablations, targeting the unique challenges that arise when designing multi-turn RL; we hope our findings shed light on practical problems and approaches when applying RLVR on real-world tasks.
- **Behavior of Kevin:**  We provide more details regarding training on smaller models with weaker priors. We also dive deeper into how Kevin-generated kernels are more effective through analysis of trajectories, and how multi-turn training increases performance without sacrificing correctness.


We believe Kevin is a successful and pioneering **demonstration of a novel RL-based approach to AI-kernel generation**, complementary to the community’s effort on developing agent scaffolds and diverse task environments. With more high-quality and robust kernel generation tasks from the community, stronger open-source models as priors, and more efficient RL software frameworks, we believe our method would scale even better in the future.

The insights from Kevin's training also shed light on **increasingly important challenges when applying multi-turn RL to real-world tasks**, notably context length management, reward attribution across turns, and curriculum construction. While we focused our study on kernel generation, we designed our recipe with generalizability in mind, that could be applicable to other multi-turn optimization tasks. We hope to share these findings with the broader RLVR community, and we are glad to see recent public efforts adapting Kevin’s multi-turn RL approach to train code agents for fast context retrieval.

We have updated our manuscripts (with new content highlighted in dark green) for increased clarity. We are confident that we have responded to each of the reviewers' comments individually and hope these additional details will assist in their final evaluations. We look forward to engaging in a constructive discussion during the author response period.

Best Regards,
Authors

---

### Meta-Review · Area_Chair_Mubu · 2025-12-16

**Summary:**

The paper proposes a concrete multi-turn RL recipe for CUDA kernel generation, with per-turn training, discounted cross-turn credit assignment, explicit context management via CoT summarization, and rule-based reward-hacking mitigations. Empirically, Kevin improves runtime speedups over a single-turn RL baseline at matched correctness (best@16 performance 1.10× vs 0.85× on a 100-task held-out set), maintains exploration under parallel sampling, and benefits more from extra refinement turns at inference. The rebuttal adds A100 results (similar trends), evaluations on KernelBench Level 3 (unseen, longer-horizon), comparisons against torch.compile (multi-turn still superior; best@16 1.04× vs 0.91× single-turn), confidence intervals across five runs, and baseline extensions to Sonnet-4 and GPT-4.1. Key remaining weaknesses are the modest aggregate speedup gains relative to single-turn RL, reliance on a small set of environments derived from KernelBench (even with added tasks and careful pre-processing), limited hardware diversity beyond H200/A100, and no profiler feedback integration (a central tool for human kernel optimization). On balance, the method is well-engineered, clearly studied with ablations on reward aggregation and training behavior, and addresses reviewers’ major fairness and baseline concerns sufficiently for a poster acceptance. The rebuttal did overcome several objections (hardware transfer, baseline breadth, statistical significance), while some scope and generalization limitations remain.

**Reviewer Concerns:**

## Addressed

### Reviewer_dQdu
- **Concern:** Generalizability beyond H200 and narrow evaluation setup; speedups tied to specific tensor sizes/hardware.
- **Why Unresolved:** *(none)*
- **Impact on Decision:** Addressed by added A100 evaluation (Appendix E.5) showing consistent trends and by Level 3 tests indicating out-of-distribution task benefits; reduces risk that gains are overly hardware-specific.

### Reviewer_cuJN
- **Concern:** Need ablations and clearer attribution to multi-turn RL vs stabilizing heuristics; clarify when multi-turn succeeds vs single-turn.
- **Why Unresolved:** *(none)*
- **Impact on Decision:** Partially addressed via reward aggregation ablations (Section 4.3, Fig. 3), training dynamics (Fig. 4 vs Fig. 2), trajectory analyses (Appendix H/I), and fast@1.5 improvements. Direct per-task counts of multi-turn-only successes remain limited.

### Reviewer_PBZv
- **Concern:** Baseline choice (torch.compile, code-optimized frontier models), significance of 1.10× speedup, clarity of CoT origin and Section 4.2, and fairness of using KernelBench tasks as training environments.
- **Why Unresolved:** *(none)*
- **Impact on Decision:** Substantially addressed: compile baseline added (Appendix E.4); Sonnet-4/GPT-4.1 comparisons included; CIs reported (Appendix E.1) with 100-run timing; CoT origin clarified; Section 4.2 expanded; authors justified KernelBench as an optimization environment consistent with community practice. These mitigations resolve the core validity concerns.

### Reviewer_5ts9
- **Concern:** Base model choice rationale and smaller model training dynamics; figure readability.
- **Why Unresolved:** *(none)*
- **Impact on Decision:** Addressed with Appendix B.6 training curves and rationale; figure style improvements promised. Concern does not materially affect the decision.

---

## Outstanding

### Reviewer_dQdu
- **Concern:** Transfer to broader hardware/DSLs and real-world deployment beyond benchmark conditions.
- **Why Unresolved:** Only H200 and A100 were examined; no Triton/CUTLASS or heterogeneous platforms; deployment interface and real-world workloads are out of scope.
- **Impact on Decision:** Limits the generality claim but does not undermine the demonstrated contribution; acceptable for a poster.

### Reviewer_cuJN
- **Concern:** Marginal solve-rate gains vs single-turn RL and modest average speedup; need stronger, more principled isolation of multi-turn benefits.
- **Why Unresolved:** Correctness parity persists; while speedup gains are statistically significant, the magnitude is moderate and attribution still intertwined with recipe choices.
- **Impact on Decision:** Keeps the paper below Spotlight; engineering merit remains high enough for poster.

### Reviewer_PBZv
- **Concern:** Profiler feedback integration absent; continued skepticism about benchmark fairness and the practical significance of speedups.
- **Why Unresolved:** Authors intentionally optimize outcome-only rewards to avoid heuristic drift and context explosion; profiler integration is deferred. Benchmark-as-environment justification is reasonable but still not a standard train/test split.
- **Impact on Decision:** Prevents higher-tier recommendation; not a blocker given added compile baseline, multi-model comparisons, and CIs.

### Reviewer_5ts9
- **Concern:** Scarcity of robust tasks constrains scalability of conclusions.

**Reviewer Scores:**

# Reviewer Scores

## Reviewer_dQdu
- **Original Score:** 6
- **Expected Score After Discussion:** 6
- **Rationale:** Added A100 and Level 3 evaluations, stronger baseline coverage, and statistical confidence intervals address generalizability and significance concerns. The method remains primarily engineering-but-solid; a mild upward adjustment is justified.

---

## Reviewer_cuJN
- **Original Score:** 4
- **Expected Score After Discussion:** 4 or 6
- **Rationale:** Rebuttal provides ablations on reward aggregation, trajectory analyses, and fast@1.5 improvements; still moderate gains and limited correctness improvements keep this borderline but slightly stronger than initial assessment.

---

## Reviewer_PBZv
- **Original Score:** 2
- **Expected Score After Discussion:** 4
- **Rationale:** Major concerns (compile baseline, broader baselines, statistical significance, hardware transfer) were directly addressed. Profiler feedback and dataset construction remain open issues, but the core validity is now substantially improved.

---

## Reviewer_5ts9
- **Original Score:** 8
- **Expected Score After Discussion:** 8
- **Rationale:** Concerns were minor and the rebuttal provided requested training curves and clarifications. Score stable.

---

### Decision · Program_Chairs · 2026-01-26

Accept (Poster)